# Neural Catalog: Scaling Species Recognition with Catalog of Life–Augmented Generation

## Abstract

Open-vocabulary species recognition is a major challenge in computer vision, particularly in ornithology, where new taxa are continually discovered. While benchmarks like CUB-200-2011 and Birdsnap have advanced fine-grained recognition under closed vocabularies, they fall short of real-world conditions. We show that current systems suffer a performance drop of over 30% in realistic open-vocabulary settings with thousands of candidate species, largely due to an increased number of visually similar and semantically ambiguous distractors. To address this, we propose Visual Re-ranking Retrieval-Augmented Generation (VR-RAG), a novel framework that links structured encyclopedic knowledge with recognition. We distill Wikipedia articles for 11,202 bird species into concise, discriminative summaries and retrieve candidates from these summaries. Unlike prior text-only approaches, VR-RAG incorporates visual information during retrieval, ensuring final predictions are both textually relevant and visually consistent with the query image. Extensive experiments across five bird classification benchmarks and two additional domains show that VR-RAG improves the average performance of the state-of-the-art Qwen2.5-VL model by 18.0%.

## 1 Introduction

The American Museum of Natural History estimates a total of 18,043 bird species Barrowclough et al. (2016). Meanwhile, the CUB-200-2011 Wah et al. (2011) dataset, a popular dataset for bird species classification, contains only 200 classes. The drastic gap questions the real-world applicability of benchmarks based on small, fixed taxonomies. Such concern goes beyond bird classification to other categories and objects. Compounding this challenge, more species are discovered annually[1]. This problem naturally becomes more challenging as new species continue to be discovered each year. The Catalogue of Life-2025 saw 48,766 newly accepted species names added COL FPC (2024). Accordingly, there is a pressing need to develop AI classification systems that can flexibly handle these challenges while providing competitive performance on existing classes.

The problem of recognizing species from an ever-increasing set is recognized as open-vocabulary recognition Wu et al. (2024b). Open-vocabulary recognition is characterized by having unseen classes during test time that the model has no idea about, and the label space itself may evolve. Learning about all the bird species is practically infeasible due to limited annotations. Therefore, we need methods that can utilize additional knowledge to recognize new species classes at test time.

A straightforward way of classifying new unseen classes in the open-vocabulary paradigm is using class descriptions. The concept of visually classifying objects from textual descriptions, along with the associated contrastive learning algorithm, can be traced back to Elhoseiny et al. (2013). And with recent and scalable **V**ision **L**anguage **M**odels (**VLM**s) like CLIP Radford et al. (2021), this task has seen rapid progress. While CLIP-like models can handle a large number and variety of classes, they underperform on open-vocabulary setups. Previous works Radford et al. (2021); Ilharco et al. (2021); Zhai et al. (2023) have evaluated the capabilities of CLIP-like models to recognize unseen classes in scenarios where the label space is restricted to species present in the dataset. In Table 1, we show that when expanding the vocabulary to $11,202$ species, the performance of CLIP drops by an average of 31.5%(35.5% across five benchmarks).

---

[1] popularmechanics.com/science/animals/a30501204/new-bird-species-discovered

While using detailed descriptions sees a similar performance pattern, it is a more promising direction for two key reasons. First, from a practical standpoint, descriptions are highly accessible: non-experts can visually describe a newly discovered species, enabling identification before a formal taxonomic name is even assigned. Second, this approach is future-proof: as new species are discovered and as we get stronger vision foundation models, these mod-

Table 1: Average accuracy drop when moving from a closed-world setting (species present in dataset) to an open-world setting with all 11,202 species.

| Model | CUB | | Birdsnap | | |
|---|---|---|---|---|---|
| | Bounded | Open | Bounded | Open | Average Drop |
| CLIP | 62.9 | 24.6 | 51.3 | 26.5 | 31.5 |
| OpenCLIP | 74.3 | 35.2 | 61.4 | 33.7 | 33.4 |
| SigLIP | 77.2 | 49.9 | 64.6 | 43.6 | 24.1 |

els can perform better with no or minimal changes by exploiting the rich detail in descriptions, overcoming the knowledge cutoffs that limit today's systems. Yet, current VLMs like CLIP Radford et al. (2021) are limited: trained on static image-caption pairs, they cannot easily integrate new knowledge and struggle with subtle morphological reasoning. We believe their shortcomings arise from a reliance on overly simplistic prompts (e.g., "a photo of a [species]") and an inability to reason about the subtle morphological differences that rich descriptions can provide.

To address these challenges, we propose Visual Re-ranking RAG (VR-RAG), a pipeline that combines MLLMs with a dynamic external knowledge base via Retrieval-Augmented Generation (RAG). Rather than training new models, VR-RAG focuses on effective integration, retrieval, and visual re-ranking to enable fine-grained, open-vocabulary species recognition. Directly applying RAG is ineffective for fine-grained species due to three challenges: (1) the sheer scale of avian biodiversity makes retrieval from comprehensive sources like Wikipedia infeasible at inference; (2) generic taxonomic descriptions often lack discriminative visual cues; (3) retrieved content may exceed MLLM context limits Yin et al. (2024).

VR-RAG uses structured encyclopedic knowledge with vision-language reasoning. First, we curate a comprehensive benchmark comprising Wikipedia articles for 11, 202 bird species, distilling each into concise, visually salient summaries using GPT-4o OpenAI (2024a) to eliminate non-discriminative text (e.g., conservation status), and extracting the visual information available and filtering them using GPT-4o. Second, we design a RAG pipeline that retrieves candidate species based on multimodal similarity. This is followed by a visual re-ranking stage that refines results using Dino-v2 Oquab et al. (2024) similarities. Finally, we augment MLLMs with the retrieved refined summaries, enabling them to reason about species identity by correlating key visual traits (e.g., "wing color", "leg color", etc.).

The principles underlying our framework are domain-agnostic and extend beyond avian classification. We demonstrate this by evaluating our method on marine species recognition on the Fish-Net Khan et al. (2023) dataset, and on our Pokémon dataset. We curate the Pokémon dataset, which includes over 1, 000 Pokémon species descriptions. Experiments across five bird benchmarks, Fish-Net Khan et al. (2023), and our Pokémon dataset demonstrate the superiority of our framework. We outperform CLIP by 35.9% on MRR@1 in the retrieval task and by 39.4% in species recognition accuracy while eliminating the need for dataset-specific retraining. In addition, we improve by 18.0% over the best MLLM model. Our contributions can be summarized as follows:

- We demonstrate that a visual re-ranking module within Retrieval Augmented Generation can substantially mitigate noisy retrieval in large-scale, fine-grained recognition. Our proposed VR-RAG improves retrieval precision (mRR@10) by 27.4% over the strongest baseline.

- We introduce an open-vocabulary benchmark for bird species recognition that spans 11, 202 species. Unlike prior datasets limited to a few hundred categories, our benchmark leverages Wikipedia-derived multimodal knowledge and GPT-4-refined summaries, enabling evaluation at realistic biodiversity scales.

- We show that integrating VR-RAG yields a substantial boost to state-of-the-art MLLMs for open-world species recognition, improving the average accuracy of the top-performing model by 18.0% across five challenging bird benchmarks.

- We further validate the cross-domain versatility of VR-RAG by applying it to FishNet and our curated Pokémon dataset, where it consistently outperforms baselines. This demonstrates that our framework generalizes beyond birds to other fine-grained recognition domains.

## 2 RELATED WORK

**Open-Vocabulary Recognition.** Early work on open-vocabulary recognition Zhao et al. (2017) introduced joint image–word embeddings. With the rise of multimodal pre-training in NLP (e.g., BERT Devlin et al. (2019)), vision–language models like CLIP Radford et al. (2021) soon emerged, later extended to detection Gu et al. (2022), segmentation Li et al. (2022), and classification Dao et al. (2023); Zhu et al. (2024). Stronger variants like OpenCLIP Ilharco et al. (2021) and SigLIP Zhai et al. (2023) achieved impressive zero-shot classification but still struggle in open-world species recognition due to limited taxonomic knowledge and dataset bias. We systematically evaluate these models and show that our approach surpasses existing baselines in open-vocabulary bird recognition.

**Species Recognition.** Species recognition is a long-standing challenge in fine-grained image classification, with birds as a central benchmark. Datasets like CUB-200-2011 Wah et al. (2011), Birdsnap Berg et al. (2014), and iNaturalist Van Horn et al. (2018) have advanced the field, but evaluations largely remain closed-set, restricted to predefined species Radford et al. (2021); Elhoseiny et al. (2013); Naeem et al. (2023). Moreover, recent work Parashar et al. (2023) shows that VLMs like CLIP are sensitive to naming conventions, while Sastry et al. (2025) further highlights their difficulty in aligning visual and taxonomic embeddings, thus proposing TaxaBind. However, recognition is open-world, where many species are unseen during training. In this work, we move to an open-vocabulary setting across five benchmarks Wah et al. (2011); Berg et al. (2014); Rokde (2023); Van Horn et al. (2018; 2015), performing recognition over the full set of species from Wikipedia.

**MultiModal Large Language Models(MLLM).** MLLMs have advanced significantly in understanding and reasoning across modalities OpenAI (2024a;b); AI (2024); DeepSeek-AI et al. (2025); Qwen et al. (2025); Jiang et al. (2024); Touvron et al. (2023). Scaling in data and models, combined with pre-training, supervised fine-tuning, and reinforcement learning from human feedback Ouyang et al. (2022), has enabled strong emergent reasoning abilities Qwen et al. (2025). However, MLLMs still degrade in long contexts Yin et al. (2024), motivating retrieval-based approaches that extend context capacity while mitigating hallucinations.

**Retrieval Augmented Generation (RAG).** RAG enhances large models by integrating external knowledge via retrieval systems Radford et al. (2021); Zhai et al. (2023); Ilharco et al. (2021); Jia et al. (2021); Johnson et al. (2021); Khattab & Zaharia (2020). The original framework by Lewis et al. (2020) jointly optimized a retriever and generator, with later work refining fusion mechanisms Izacard & Grave (2021) and extending RAG to long-form reasoning, multi-hop QA, and hallucination mitigation Mallen et al. (2023); Asai et al. (2023). Vision-centric extensions include MuRAG Chen et al. (2022), which uses an image-text memory bank, MIRAGE Wu et al. (2025) employs a CLIP-based retriever, and REVEAL Hu et al. (2023) leverages multimodal graphs for reasoning. We propose VR-RAG, a two-stage framework that fuses multiple vision encoders and applies a re-ranker to refine top-$k$ candidates, achieving superior performance over existing methods.

## 3 DATASET AND BENCHMARK

Our benchmark comprises 11,202 species, each paired with a corresponding Wikipedia article, a concise summary describing the species, and representative images referred to as anchor images.

**Data Collection.** We begin by compiling a comprehensive list of all bird species available on Wikipedia [2]. For each species, we retrieve all information from its respective Wikipedia page, both textual and visual. Species without dedicated pages are excluded from our dataset. This process results in a curated set of $11,202$ bird species, each accompanied by its Wikipedia-sourced description and visual information, forming a rich knowledge base for our open-vocabulary recognition task.

**Summary Generation and Visual Refinement.** Wikipedia articles often contain extraneous information, such as name origins and historical context, which are useless for visual discrimination. This irrelevant content can degrade the performance by increasing the context length without adding

---

[2]https://en.wikipedia.org/wiki/List_of_birds_by_common_name

meaningful visual cues. To address this, we employ GPT-4o OpenAI (2024a) to refine the Wikipedia articles by generating concise summaries focused on attributes relevant to distinguishing bird species visually. We do so by prompting GPT-4o OpenAI (2024a) with the species Wikipedia article and prompting it with: *Summarize the following information about the bird species "species name" into a concise paragraph. Focus on the key physical characteristics that distinguish this species from other similar bird species. Highlight features like size, beak shape, plumage color patterns, wing shape, and any other unique traits. The summary should be useful for someone trying to identify the bird species from a photograph.* This reduces the average word count per species from 552 to 127.

The visual content extracted from Wikipedia articles is often heterogeneous, including not only photographs of the species but also non-target materials such as habitat maps, conservation park logos, and scientific illustrations. To filter the authentic photographs, each image was fed to GPT-4o and evaluated with the direct prompt:"Is there a bird present in the Image? Reply only with Yes or No". Only images that received a 'Yes' response were retained. The retained images serve as representative "anchor images" for each species. After filtering, each species had an average of three anchor images associated with it.

**Pokémon dataset.** We built our Pokémon dataset through a multi-stage pipeline. We began with the canonical list of all Pokémon species from Pokémondb text list. For each species, we used GPT-4o to generate a concise, descriptive summary highlighting key visual characteristics from Wikipedia in a similar fashion as done for birds. Image collection was performed by scraping results from Google Search and multiple Kaggle datasets based on Pokémon. However, we noticed the scraped images often contained multiple Pokémons or were of the wrong species. To ensure overall image quality and consistency, all scraped images were manually verified against a representative image for each species. The image was selected from Pokémondb Pokedex page. All the inaccurate samples were discarded. This curation process yielded a final test set of 11,216 images spanning 248 species.

**Description Quality.** We evaluate the description quality by performing human evaluation in two ways: 1) The users are shown four images of the corresponding species along with the description, and they are asked to rate the description on a 1-5 scale, with 1 indicating "not helpful at all" and 5 indicating "very helpful" for identifying the species. This was done for 10% of the bird species descriptions and all 248 descriptions from the Pokémondataset. Each description was rated by three human annotators. Bird descriptions received a mean score of 4.3, a median of 4.0, and a mode of 5.0, while Pokémondescriptions received a mean score of 4.5, a median of 5.0, and a mode of 5.0. For consistency, we calculated inter-annotator reliability using Quadratic Weighted Agreement (QWA). Across both tasks, we achieved a QWA of 90.6% for bird descriptions and 92.2% for Pokémon descriptions. The complete breakdown of the calculation of the reliability scores can be found in appendix L. 2) We conduct a matching test, where the users are shown a description and five images, and they are asked to choose the image that is most matching. To ensure the task is fine-grained, we conducted this test on the 200 species from the cub dataset. We ensure the correct image is always present in the five images, and the remaining four are chosen as the closest to the description in the clip space. On average, the users spent almost one minute on each description and achieved 94% accuracy. For the remaining 6%, 4% cases were where the users could not converge to one single option, and the remaining 2% were wrong. We discuss and showcase qualitative samples for all the cases with provided explanation in appendix L.

## 4 PRELIMINARIES

This section introduces the problem setting and provides the necessary background knowledge.

### 4.1 TASK FORMULATION

We focus on *open-vocabulary species recognition*, which aims to determine the species of a given image. Unlike traditional classification tasks, where the label space is restricted to species present in the training database Wu et al. (2024a), our task considers an open and ever-expanding textual label space, making the problem inherently cross-modal.

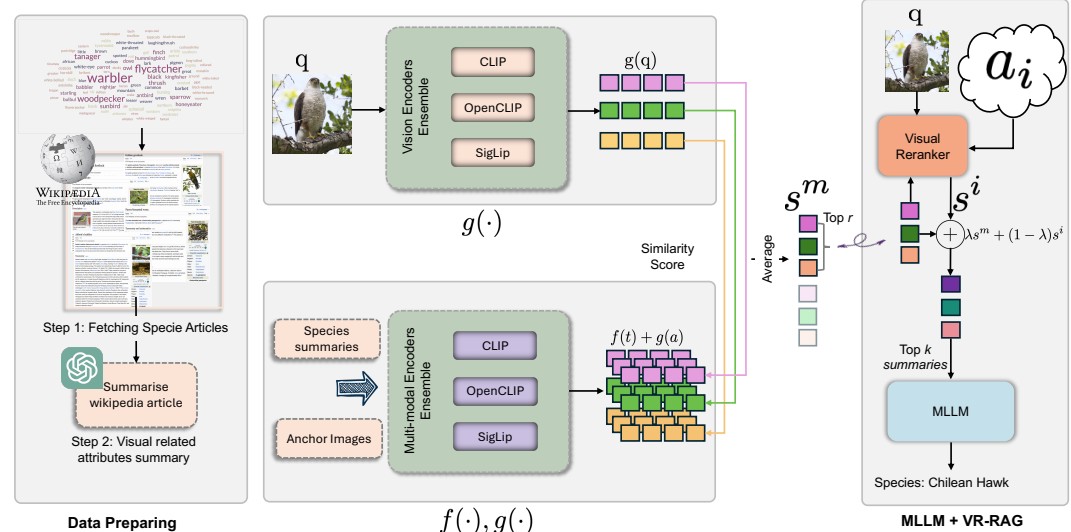

Figure 1: **The VR-RAG pipeline**. Left: Data extraction, both textual and visual, from Wikipedia articles. Middle: Similarity calculation with the query image using an ensemble of multi-modal encoders. Right: Re-ranker module using anchor images for final similarity score calculation.

## 4.2 VISION-LANGUAGE MODELS (VLMS)

Vision-Language Models (VLMs) like CLIP Radford et al. (2021) map text $y$ and images $x$ into a shared representation space using a text encoder $f(\cdot)$ and vision encoder $g(\cdot)$. VLMs are trained on large-scale image-text datasets, such as WIT-400M Schuhmann et al. (2021) and LAION Schuhmann et al. (2022), using contrastive learning, where text representations $f(y)$ and image representations $g(x)$ of corresponding caption-image pairs are aligned. While effective for many vision tasks Zhang et al. (2024b), VLMs alone are insufficient for our task, as they lack the reasoning capabilities to differentiate between similar species Patel et al. (2024); Abbasi et al. (2025). Therefore, additional reasoning mechanisms are needed to improve recognition performance.

## 4.3 MULTI-MODAL LARGE LANGUAGE MODELS (MLLMS)

MLLMs such as Qwen et al. (2025); Chen et al. (2024); Yin et al. (2024) are trained on massive multimodal datasets, enabling them to handle various vision and language tasks. These models possess reasoning capabilities and can integrate information from both textual and visual inputs. However, they are still prone to hallucinations when dealing with large contexts Jin et al. (2025). To mitigate this, RAG has been proposed to provide up-to-date information as context.. However, the performance gains from RAG are highly dependent on the design of an effective multimodal retrieval mechanism.

## 5 METHOD

Given a query image $q$ and a set of $\mathcal{N}$ species summaries and associated anchor images $a_i$, our approach consists of a retrieval module followed by a reranker module and then an MLLM module. First, we retrieve the top $r$ species summary chunks using our retrieval module. These chunks are then re-ranked using a visual re-ranker module to obtain the top $k$ chunks. The chunks are then mapped to their respective species summaries, which are subsequently provided to the MLLM along with the query image $q$ to determine the species present in the query image.

### 5.1 RETRIEVAL

To adhere to the context length limitations of our models, we first segment each of the $\mathcal{N}$ species summaries into smaller text chunks. For each species, its set of $K$ anchor images $\{a_i\}_{i=1}^{K}$ are

processed by a visual encoder $g(\cdot)$, and their embeddings are averaged to create a single visual representation. Concurrently, each associated text chunk, $t$, is processed by a text encoder $f(\cdot)$ to produce a text embedding. Each chunk is represented by its multi-modal representation, $m$ by:

$$m = \frac{f(t) + \frac{1}{K}\sum_{i=1}^{K} g(a_i)}{2} \tag{1}$$

We then compute the similarity between a given query image $q$ and each multi-modal representation $m$. This multi-modal similarity $s^m$, is calculated as the dot product between the query image's embedding $g(q)$ and the multi-modal embedding $m$, $s^m = g(q)^\top m$. All embeddings are $l_2$-normalised.

Ensemble methods have been shown to improve performance across various machine learning tasks Hansen & Salamon (1990). We observe a similar trend with VLMs, leading us to employ an ensemble for improved retrieval. Specifically, we combine the similarity scores from CLIP, OpenCLIP, and SigLIP by averaging their similarity scores to obtain a final ranking. This ensemble approach effectively leverages the unique strengths of each VLM, resulting in better retrieval performance than any single model could achieve on its own. This is shown in the central part of fig. 1

## 5.2 RERANKER

After retrieving the top $r$ chunks, we refine the selection further by re-ranking them to obtain the top $k$ chunks. This re-ranking step utilizes both textual relevance and visual similarity. For each species represented in the top $r$ chunks, we sample the anchor images. We then calculate the intra-modal similarity between the anchors $a_i$ and query $q$. For our vision encoder $h$, we use Dino-v2 Oquab et al. (2024), which has demonstrated strong performance in visual understanding-related tasks Tong et al. (2024b;a); Shen et al. (2024). The final similarity score for each chunk is obtained by a combination of intra-modal and the previously calculated multi-modal similarity weighted by $\lambda$, as defined in eq. (2). The top $k$ chunks, after re-ranking, are selected for the next stage. A visual approach to our re-ranker module is presented in fig. 7.

$$s^i = h(q)^\top \left( \frac{1}{K}\sum_{i=1}^{K} h(a_i) \right) \tag{2}$$
$$s = \lambda s^m + (1 - \lambda)s^i$$

## 5.3 MLLM

Instead of passing only the top-$k$ retrieved chunks, we input the full summaries of the $k$ species associated with the top-$k$ ranked chunks to the MLLM, along with the query image $q$. This is feasible because our summaries are concise and discriminative, unlike lengthy Wikipedia articles that often are filled with non-discriminative details. As a result, even when $k$ increases (e.g., 5, 10, or 15), the MLLM can process all candidate summaries within its context window and make a more informed decision about the species in $q$ without being overwhelmed by irrelevant information. This is ablated in table 7 where we show that providing our refined summary works best by improving 4.0% over raw Wikipedia articles and 12.4% over using only retrieved chunks as context.

## 6 EXPERIMENTS AND RESULTS

Below, we outline our experimental setup, detailing the datasets used, evaluation metrics, and baseline methods. We then present our results, accompanied by an ablation study, to demonstrate the effectiveness of each component.

### 6.1 EXPERIMENTAL SETUP

**Datasets.** We evaluate our approach on five bird classification datasets. **CUB-200-2011** Wah et al. (2011) consists of 11,788 images of 200 bird species from North and South America, with 5,794 images used for testing. **Birdsnap** Berg et al. (2014) is a larger dataset of North American birds, containing 49,829 images spanning 500 species, with 2,443 images allocated for testing. **Indian**

**Birds** Rokde (2023) includes 25 species with a test set of 7,499 images. **iNaturalist** Van Horn et al. (2018) is a large-scale biodiversity dataset covering thousands of plant and animal species, designed for real-world fine-grained classification challenges. We extract all images of the 'Aves' class from the validation set, resulting in 13,230 images spanning 1,323 bird species. **NaBirds** Van Horn et al. (2015) is a fine-grained bird classification dataset focused on North American species. It comprises 48,562 annotated images across 555 visual categories, which correspond to 400 unique bird species, with 24,633 images for testing. At test time, all 11,202 collected species summaries are ranked for each image, aligning the evaluation with an open-vocabulary setting.

We also evaluate VR-RAG on two additional datasets. The first contains marine species from the **FishNet** Khan et al. (2023) dataset, where we follow the original split strategy for species classification. To align with our open-vocabulary setting, each query image is compared against 17,393 candidate descriptions sourced from FishNet. For the **Pokémon** dataset, we evaluate on the full set of 248 species using 1,024 descriptions.

**Metrics.** For retrieval evaluation, we use mean reciprocal rank (mRR), which is the average inverse rank of the first retrieved ground truth image. We report mRR@1, mRR@5, and mRR@10 to evaluate the ranking of the first correct among the top retrieved chunks. For recognition, we report the percentage of correctly classified examples to measure performance.

**Baselines.** In our experiment, we evaluate several VLMs for retrieval and both VLMs and MLLMs for recognition tasks. For the retrieval problem, we use CLIP Radford et al. (2021), OpenCLIP Ilharco et al. (2021), and SigLIP Zhai et al. (2023), all of which employ the ViT-L14 as the backbone. We use BioClip Stevens et al. (2024), which employs the ViT-B16 backbone.

For the recognition task, we evaluate the same VLMs used for retrieval. We follow the standard protocol of calculating similarity and selecting the highest-rated chunk. The chunk is mapped to the species it belongs to evaluate CLIP-like models. Although these models are trained to match object categories with images, our setting involves working with summaries. Therefore, we compute the similarity between the query image and summary chunks for a fair evaluation.

We assess multiple MLLMs, including Qwen2-VL-7B Instruct from the Qwen2-VL Yang et al. (2024) suite, Qwen2.5-VL-7B Instruct from Qwen2.5-VL Qwen et al. (2025), Gemma-3n-e4b-it Team et al. (2025), Mini-CPM-V-2.9 Yao et al. (2024) and InternVL3-8B from Intern-VL Zhu et al. (2025). We assess these models in two settings, one by directly asking them the species name given a query image, and the second by providing summaries for the top $k$ species retrieved via VR-RAG as context. We select $r$ as 100, $k$ as 5, and $\lambda$ as 0.7 for our experiments.

## 6.2 RESULTS

**Retrieval.** We present the retrieval results in the table 2. Across all five benchmarks, VR-RAG outperforms the baseline methods. It achieves the highest scores for all levels (1, 5, 10) on the mRR metric. On average, VR-RAG improves upon the best baseline (SigLIP Zhai et al. (2023)) by 32.8% on MRR@1, 27.8% on MRR@5, and 27.4% on MRR@10. These improvements highlight the strength of our approach, which combines three vision-language encoders and a visual re-ranker.

Table 2: Retrieval Results: We compare VR-RAG with other cross-modal(text-to-image) retrieval methods across the five benchmarks. VR-RAG consistently outperforms the baseline models on mRR@1, mRR@5, and mRR@10 metrics. The best results are highlighted in bold text.

| | Birdsnap | | | CUB | | | iNaturalist Birds | | |
|---|---|---|---|---|---|---|---|---|---|
| | mRR@1 | mRR@5 | mRR@10 | mRR@1 | mRR@5 | mRR@10 | mRR@1 | mRR@5 | mRR@10 |
| BioCLIP | 19.4 | 26.5 | 27.7 | 24.3 | 31.2 | 32.4 | 13.5 | 19.2 | 20.2 |
| CLIP | 17.2 | 25.1 | 26.5 | 16.1 | 24.1 | 25.6 | 7.7 | 12.3 | 13.4 |
| OpenCLIP | 15.7 | 24.8 | 26.3 | 15.2 | 23.9 | 25.4 | 8.5 | 13.7 | 14.9 |
| SigLIP | 18.8 | 27.9 | 29.3 | 20.1 | 29.6 | 31.1 | 11.3 | 17.2 | 18.3 |
| **VR-RAG(ours)** | **48.9** | **52.6** | **53.8** | **58.0** | **62.3** | **63.1** | **34.8** | **38.7** | **39.8** |
| | Indian Birds | | | NABirds | | | Average | | |
| | mRR@1 | mRR@5 | mRR@10 | mRR@1 | mRR@5 | mRR@10 | mRR@1 | mRR@5 | mRR@10 |
| BioCLIP | 23.0 | 29.2 | 30.0 | 21.8 | 29.1 | 30.3 | 20.4 | 27.0 | 28.1 |
| CLIP | 17.6 | 25.9 | 27.2 | 18.2 | 26.4 | 27.9 | 15.4 | 22.8 | 24.1 |
| OpenCLIP | 10.2 | 16.8 | 18.3 | 15.5 | 23.9 | 25.6 | 13.0 | 20.6 | 22.1 |
| SigLIP | 21.9 | 30.4 | 31.9 | 20.5 | 29.6 | 31.1 | 18.5 | 26.9 | 28.3 |
| **VR-RAG(ours)** | **68.1** | **72.9** | **73.6** | **52.0** | **56.6** | **57.6** | **52.3** | **56.6** | **57.6** |

**Recognition.** In table 3, we present the recognition results across all five benchmarks. A consistent performance improvement is observed across MLLMs when augmented with VR-RAG. In contrast, direct RAG negatively impacts QWEN2.5-VL, which we attribute to the inferior retrieval capabilities of CLIP compared to our VR-RAG module. Specifically, VR-RAG enhances QWEN2-VL by 22.3%, and QWEN2.5-VL by 18.0%. We also evaluated GPT-4o on the iNat-Birds dataset, as it has the most extensive species coverage among the five datasets. In a direct zero-shot setting, GPT-4o achieved 23.9% accuracy, outperforming all open-source models tested. When augmented with our VR-RAg framework, its performance

Table 3: Classification Accuracy: We evaluate open-source MLLMs directly and integrate them with our VR-RAG pipeline. We also show results for various VLMs. Direct RAG uses CLIP Radford et al. (2021) as the RAG module.

| | Birdsnap | CUB | iNat. | Ind. Birds | NABirds | Average |
|---|---|---|---|---|---|---|
| BioCLIP Stevens et al. (2024) | 19.4 | 24.3 | 13.5 | 23.0 | 21.8 | 20.4 |
| CLIP Radford et al. (2021) | 17.2 | 16.1 | 7.7 | 17.6 | 18.2 | 15.4 |
| OpenCLIP Ilharco et al. (2021) | 15.7 | 15.2 | 8.5 | 10.2 | 15.5 | 13.0 |
| SigLIP Zhai et al. (2023) | 18.8 | 20.1 | 11.3 | 21.9 | 20.5 | 18.5 |
| InternVL-3 Zhu et al. (2025) | 12.7 | 14.9 | 4.0 | 0.5 | 14.4 | 9.3 |
| MiniCPM-V-2.6 Yao et al. (2024) | 9.4 | 8.0 | 2.8 | 1.8 | 10.8 | 6.6 |
| Qwen2VL Yang et al. (2024) | 34.3 | 37.2 | 14.7 | 24.0 | 42.7 | 30.6 |
| Gemma-3n Yang et al. (2024) | 17.2 | 19.8 | 8.5 | 15.6 | 22.8 | 16.8 |
| Qwen2.5VL Qwen et al. (2025) | 39.3 | 44.7 | 17.9 | 33.1 | 49.2 | 36.8 |
| Qwen 2.5VL + direct RAG | 29.5 | 29.2 | 13.2 | 29.5 | 31.7 | 26.6 |
| Qwen 2VL + VR-RAG (ours) | 49.8 | 57.3 | 34.1 | 70.2 | 55.5 | 53.4 |
| Qwen 2.5VL + VR-RAG (ours) | **51.9** | **60.3** | **35.5** | **72.2** | **56.3** | **55.2** |

increased substantially to 36.0%. Notably, this is only slightly higher than the performance of the augmented Qwen2.5-VL, highlighting the strength of open-source models that are significantly smaller than GPT-4o. Notably, VLMs such as CLIP Radford et al. (2021) perform poorly due to their inability to effectively reason over detailed summaries Patel et al. (2024); Abbasi et al. (2025) and their limited context window Zhang et al. (2024a). VR-RAG, on the other hand, provides a flexible and easily integrable framework for species recognition, making it particularly advantageous in dynamic scenarios where species may be newly discovered or become extinct. This eliminates the need for full model retraining, ensuring adaptability in open-world settings. Results for the remaining MLLMs with VR-RAG are presented in appendix A.

## 6.3 ADDITIONAL DOMAINS

We present results for FishNet and the Pokémon dataset in table 4, highlighting the best-performing MLLM and VLM. Both VLMs and MLLMs perform poorly on Fish-Net, likely due to the lack of marine species knowledge and the extremely fine-grained nature of the task, which involves a large vocabulary of over 17,000 species.

Table 4: Classification accuracy for the FishNet and Pokémon datasets.

| | FishNet | Pokémon |
|---|---|---|
| SigLIP Zhai et al. (2023) | 3.1 | 75.5 |
| Qwen2.5VL Qwen et al. (2025) | 5.0 | 29.4 |
| Qwen 2.5VL + direct RAG | 4.5 | 60.5 |
| Qwen 2.5VL + VR-RAG (ours) | **18.7** | **86.3** |

While these results are unsurprising, our approach still improves upon the best MLLM by 12.9%. For the Pokémon dataset, SigLIP outperforms QWEN2.5-VL by a huge gap(46.1%). But, when augmented by VR-RAG, it improves by 56.2%, clearly showcasing that VR-RAG is highly effective across domains. Detailed evaluations on both datasets are provided in appendix B.

## 6.4 ADDITIONAL BASELINES

We compare our method with additional baselines Menon & Vondrick (2023); Liu et al. (2023) by substituting their retrieval components into our pipeline. Classification by description Menon & Vondrick (2023) uses GPT-generated visual descriptors, while REACT Liu et al. (2023) adapts CLIP using a small set of web images for strong zero-/few-shot performance. As shown in Table 5, both struggle in true open-vocabulary settings. Incorporating our visual re-ranking stage significantly improves their performance, demonstrating that VR-RAG integrates easily with diverse retrieval methods and can further benefit from future advances in vision backbones or retrieval pipelines.

Table 5: Baseline comparison.

| | Birdsnap | CUB | iNat. | Ind. Birds | NABirds | Average |
|---|---|---|---|---|---|---|
| Menon & Vondrick (2023) | 34.7 | 36.2 | 17.6 | 26.1 | 38.3 | 30.6 |
| Menon & Vondrick (2023)+VR-RAG | 46.9 | 55.8 | 30.8 | 60.2 | 53.1 | 49.4 |
| Liu et al. (2023) | 28.9 | 30.2 | 16.7 | 24.9 | 32.3 | 26.6 |
| Liu et al. (2023)+VR-RAG | 42.3 | 52.5 | 28.3 | 52.1 | 45.2 | 44.1 |

## 6.5 ABLATION

**Retrieval Components.** The ablation study in table 6 illustrates the contribution of each component in the VR-RAG pipeline. CLIP alone performs poorly across all benchmarks, but incremental additions of OpenCLIP and SigLIP lead to consistent improvements. The best encoder performance

Table 6: Ablation study: We evaluate each of VR-RAG module's impact for the MRR@1, MRR@5, and MRR@10 on CUB-200-2011 Wah et al. (2011), Birdsnap Berg et al. (2014), and Indian Birds Rokde (2023). The best results are highlighted in bold text for each column.

| CLIP | OpenCLIP | SigLIP | Multimodal | Re-Ranker | Birdsnap Berg et al. (2014) | | | CUB Wah et al. (2011) | | |
|---|---|---|---|---|---|---|---|---|---|---|
| | | | | | mRR@1 | mRR@5 | mRR@10 | mRR@1 | mRR@5 | mRR@10 |
| ✓ | ✗ | ✗ | ✗ | ✗ | 17.2 | 25.1 | 26.5 | 16.1 | 24.1 | 25.6 |
| ✓ | ✓ | ✗ | ✗ | ✗ | 25.5 | 35.5 | 36.9 | 25.5 | 36.0 | 37.4 |
| ✓ | ✓ | ✓ | ✗ | ✗ | 28.8 | 39.2 | 40.5 | 30.0 | 40.9 | 42.4 |
| ✓ | ✓ | ✓ | ✓ | ✗ | 40.6 | 47.3 | 48.5 | 40.9 | 42.4 | 48.6 |
| ✓ | ✓ | ✓ | ✓ | ✓ | **48.9** | **52.6** | **53.8** | **58.0** | **62.3** | **63.1** |

is achieved when all three are combined. Incorporating multi-modal embeddings, comprising textual summaries and visual anchors, further boosts performance across all datasets. Finally, employing Dino-V2 to rerank the retrieved chunks results in significant gains across all mRR levels. Overall, the study confirms that every component of VR-RAG is essential to its effectiveness. An ablation of MLLM performance when attached to each of the retrieval pipelines from table 6 is presented in appendix D, and ablations for VLMs and re-rankers are presented in appendix C.

**Top-$k$ Selection.** In fig. 2, we analyze the impact of increasing the number of top-k species candidates from $k = 5$ to $k = 15$ on performance. The data reveal a consistent inverse relationship between context length and accuracy, suggesting that the models struggle to identify the relevant information within a longer candidate list.

For Qwen2.5-VL, average accuracy steadily declines from 54.8% at $k = 5$ to 54.3% at $k = 10$, and further to 52.8% at $k = 15$. This drop of two percentage points confirms that the expanded context with more options hinders the model's reasoning. This decline in performance indicates that by providing more candidate species, the models appear to be overwhelmed by the increased number of options, making it more challenging to focus on the sub-

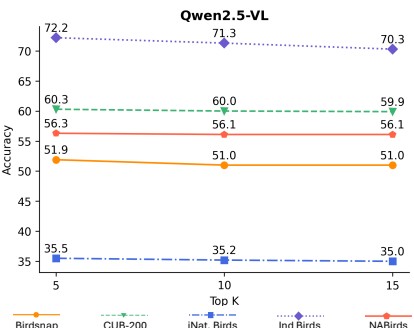

Figure 2: Impact of varying top-k candidates(5 to 15) fed to Qwen2.5-VL.

tle, fine-grained details needed for accurate species classification. This highlights a key limitation in these vision-language models for this specific task: their inability to effectively filter out redundant or confusing information when presented with a large set of candidates. It also emphasizes the importance of providing a concise, high-quality context rather than a large volume of raw data. Despite the drop, it is noteworthy that even at $k = 15$, the performance(52.8%) of Qwen2.5-VL is superior to providing raw Wikipedia articles(51.2%).

**Varying $\lambda$.** The final re-ranked similarity score is a weighted average of the multi-modal and the intra-modal similarity from DinoV2, with the balance controlled by the parameter $\lambda$. As shown in fig. 3, the optimal performance is achieved at $\lambda = 0.7$. This indicates that slightly favoring the multi-modal similarity yields the best results, while relying too heavily on an individual similarity (as $\lambda$ approaches 0 or 1) leads to suboptimal performance.

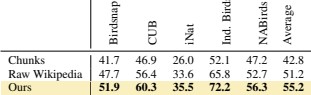

Figure 3: Impact of varying $\lambda$.

**Ablating summary refinement pipeline.** As shown in table 7, QWEN2.5-VL performs poorly when given only top-ranked chunks, as they lack sufficient detail to distinguish the query image. Using the full Wikipedia article improves accuracy but adds large amounts of irrelevant text, increasing noise and processing cost. In contrast, our refined summaries strike a balance by retaining only discriminative details, enabling better reasoning and yielding the strongest performance.

Table 7: Studying the impact of different types of context fed to MLLM.

| | Birdsnap | CUB | iNat | Ind. Birds | NABirds | Average |
|---|---|---|---|---|---|---|
| Chunks | 41.7 | 46.9 | 26.0 | 52.1 | 47.2 | 42.8 |
| Raw Wikipedia | 47.7 | 56.4 | 33.6 | 65.8 | 52.7 | 51.2 |
| Ours | **51.9** | **60.3** | **35.5** | **72.2** | **56.3** | **55.2** |

**Noisy Database.** We study the impact of database noise on retrieval quality. To evaluate this, instead of using our refined summaries, we perform retrieval using the

Table 8: Studying retrieval robustness.

| | Birdsnap | | | CUB | | |
|---|---|---|---|---|---|---|
| | mRR@1 | mRR@5 | mRR@10 | mRR@1 | mRR@5 | mRR@10 |
| VR-RAG(noisy) | 47.1 | 48.6 | 49.0 | 56.8 | 58.5 | 59.0 |
| VR-RAG | 48.9 | 52.6 | 53.8 | 58.0 | 62.3 | 63.1 |

full Wikipedia corpus. As shown in Table 8, although performance drops slightly, it remains significantly higher than that of individual models such as CLIP in Table 1. This demonstrates that our approach is robust to noise in the database and continues to perform well even under such conditions.

**Ablating prompt robustness.** To assess robustness, we evaluated our method using five different prompts and report both the best and average performance in table 9. Although some prompts perform slightly better than others, the overall variance is small, indicating that the pipeline is largely insensitive to prompt choice.

Table 9: Studying prompt robustness.

|  | Birdsnap | CUB | iNat | Ind. Birds | NABirds | Average |
|---|---|---|---|---|---|---|
| Average | 49.6 | 58.8 | 34.2 | 68.8 | 55.0 | 53.3 |
| Best | 51.9 | 60.3 | 35.5 | 72.2 | 56.3 | 55.2 |

### 6.6 COMPUTATIONAL OVERHEAD

We analyze the computational efficiency of our method relative to several widely used baselines. Figure 4 presents a three-way trade-off between retrieval performance (mrr@1), inference latency, and peak GPU memory consumption. We observe that lightweight encoders such as CLIP, OpenCLIP, and SigLIP offer low latency (0.15–0.18s) and low peak memory usage (1.3–3.1 GB), but their retrieval performance remains significantly lower (13–18 mrr@1). Combining these three encoders (CLIP, OpenCLIP, SigLIP) substantially improves accuracy (27 mRR@1) at the cost of increased latency (0.25s) and peak GPU memory usage (5.9 GB). Introducing multimodal embeddings by using visual information as described

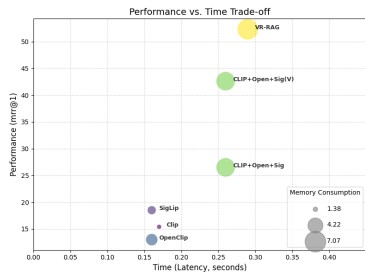

Figure 4: Performance vs. Latency vs. Memory Trade-off

in eq. (1), yields a further boost to 43 mrr@1 with no change in memory footprint and only a small latency increase. Our full system, VR-RAG, achieves the highest retrieval performance (52 mrr@1) while maintaining latency under 0.29s and a peak memory cost of $\approx$7 GB. VR-RAG offers a significant performance gain relative to the computational overhead.

These results demonstrate that although our approach introduces moderate additional cost compared to basic encoders, it delivers disproportionately higher accuracy. From a systems perspective, VR-RAG offers a desired balance between effectiveness, latency, and memory, making it practical for deployment in high-accuracy scientific monitoring or large-scale retrieval systems.

## 7 CONCLUSION

In this work, we focus on the task of open-vocabulary species recognition and demonstrate the limitations of current Vision-Language Models (VLMs) in this setting. To address these challenges, we propose VR-RAG, a multimodal re-ranking retrieval-augmented generation (RAG) framework that significantly improves retrieval performance over a large pool of species summaries.

Additionally, we collect and refine Wikipedia summaries for 11, 202 bird species, distilling them into short, discriminative descriptions. We curate a Pokémon dataset with descriptions for over 1, 000 Pokémons. Experimental results show that augmenting VR-RAG with MLLMs leads to superior performance across all five avian, one marine, and one Pokémon benchmark.

Future directions include enabling models to identify when species summaries are insufficient and autonomously retrieve supplementary information from external sources to support fine-grained identification. However, our approach depends on the availability and quality of public textual resources, and while birds offer near-complete coverage, this may not be the case for other underexplored taxa.

## 8 ETHICS

The primary goal of this research is to advance AI for a positive societal impact, specifically in biodiversity conservation. All the textual data sourced from encyclopedic resources like Wikipedia will be made publicly available, with an appropriate license. The dataset contains images of animal

species and does not involve human subjects, thus presenting no personal data privacy concerns. We believe the potential for misuse of this technology is low, as its primary application is intended for scientific research and environmental monitoring.

## 9 REPRODUCIBILITY

We are committed to ensuring reproducibility of our work. All datasets, including the curated Wikipedia-based benchmark and the Pokémon dataset, will be made publicly available under appropriate licenses. We specify hyperparameters and architectures for evaluation in the main text and supplementary material. Our codebase, data loaders, and evaluation scripts will be released on GitHub. Random seeds are fixed in all experiments. Together, these steps ensure that our results can be independently verified and extended by the community.

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

## A ADDITIONAL RESULTS WITH VR-RAG.

In this section, we present results for the remaining MLLMs from table 10 when integrated with VR-RAG, and compare them against direct-RAG, which uses CLIP as the retrieval module. As shown in table 10, VR-RAG consistently delivers clear performance gains across all five benchmarks.

Table 10: Classification Results for remaining MLLMs

| | Birdsnap | CUB | iNat. | Ind. Birds | NABirds | Average |
|---|---|---|---|---|---|---|
| InternVL-3 Zhu et al. (2025) | 12.7 | 14.9 | 4.0 | 0.5 | 14.4 | 9.3 |
| InternVL-3 + Direct-RAG | 27.0 | 27.0 | 12.6 | 25.8 | 28.3 | 24.1 |
| InternVL-3 + VR-RAG | 42.0 | 46.6 | 27.8 | 48.9 | 45.3 | 42.1 |
| MiniCPM-V-2.6 Yao et al. (2024) | 9.4 | 8.0 | 2.8 | 1.8 | 10.8 | 6.6 |
| MiniCPM-V-2.6 + Direct-RAG | 22.7 | 24.1 | 10.5 | 23.3 | 26.6 | 21.4 |
| MiniCPM-V-2.6 + VR-RAG | 36.0 | 40.8 | 24.6 | 44.7 | 40.8 | 37.4 |
| Gemma-3n Yang et al. (2024) | 17.2 | 19.8 | 8.5 | 15.6 | 22.8 | 16.8 |
| Gemma-3n + Direct-RAG | 25.2 | 25.5 | 12.0 | 27.6 | 27.5 | 23.6 |
| Gemma-3n + VR-RAG | 44.8 | 52.7 | 32.2 | 61.1 | 48.3 | 47.8 |

## B COMPLETE EVALUATION FOR FISHNET AND POKÉMON.

In table 11, we present the performance of all models on the FishNet and the Pokémon datasets. All models struggle with the highly fine-grained and diverse FishNet dataset. Even in this challenging setting, the application of VR-RAG significantly improves the performance of MLLMs.

On the Pokémon dataset, all Vision-Language Models except BioCLIP achieve superior performance compared to MLLMs. BioCLIP, having been exclusively trained on natural data from the Tree of Life Stevens et al. (2024); Hinchliff et al. (2015), likely lacks exposure to non-natural domains, which explains its poor performance on this dataset. In contrast, the re-

Table 11: Complete classification results for the FishNet and the Pokémon dataset.

| | FishNet | Pokémon |
|---|---|---|
| BioCLIP Stevens et al. (2024) | 2.7 | 0.2 |
| CLIP Radford et al. (2021) | 2.8 | 60.5 |
| OpenCLIP Ilharco et al. (2021) | 1.2 | 56.2 |
| SigLIP Zhai et al. (2023) | 3.1 | 75.5 |
| InternVL-3 Zhu et al. (2025) | 5.6 | 12.6 |
| MiniCPM-V-2.6 Yao et al. (2024) | 1.3 | 17.1 |
| Qwen2VL Yang et al. (2024) | 3.9 | 30.1 |
| Gemma-3n Yang et al. (2024) | 4.7 | 30.8 |
| Qwen2.5VL Qwen et al. (2025) | 5.0 | 29.4 |
| Qwen 2.5VL + direct RAG | 4.5 | 60.5 |
| Qwen 2VL + VR-RAG (ours) | 15.3 | 81.6 |
| Qwen 2.5VL + VR-RAG (ours) | **18.7** | **86.3** |

maining VLMs perform exceptionally well, outperforming all MLLMs. However, when Qwen2.5-VL is augmented with VR-RAG, it achieves the best overall performance, demonstrating the effectiveness of our retrieval-augmented approach even in less-specialized domains compared to VLMs.

## C EVALUATION WITH VLM USING BOTH ENCODERS.

In table 12, we evaluate the performance of our VR-RAG encoders, examining how their multimodal representations, combining both text and visual encoders, enhance retrieval across all five benchmarks. The results demonstrate that this fusion of modalities consistently improves retrieval accuracy.

Furthermore, in table 13, we present a performance comparison when different visual encoders are used as re-rankers. We test the visual encoders from the VR-RAG pipeline, DinoV2 Oquab et al. (2024) and DeiT-III Touvron et al. (2022). The data clearly shows that DinoV2 significantly outperforms not only DeiT-

Table 12: Retrieval results for the VLMs in VR-RAG when combining both text and visual representations, and two visual encoders. We report results for mRR@5

| | Birdsnap | CUB | iNat. | Ind. Birds | NABirds | Average |
|---|---|---|---|---|---|---|
| $CLIP_t$ | 25.1 | 24.1 | 12.3 | 25.9 | 26.4 | 22.8 |
| $CLIP_v$ | 22.9 | 31.8 | 15.8 | 31.5 | 27.0 | 25.8 |
| $CLIP_{t+v}$ | 33.3 | 41.5 | 21.3 | 47.0 | 37.8 | 36.2 |
| $OpenCLIP_t$ | 24.8 | 23.9 | 13.7 | 16.8 | 23.9 | 20.6 |
| $OpenCLIP_v$ | 33.1 | 42.6 | 21.1 | 44.3 | 37.4 | 35.7 |
| $OpenCLIP_{t+v}$ | 43.7 | 50.9 | 26.9 | 50.3 | 46.9 | 43.7 |
| $SigLIP_t$ | 27.9 | 29.6 | 17.2 | 30.4 | 29.6 | 26.9 |
| $SigLIP_v$ | 35.6 | 44.5 | 22.9 | 49.3 | 40.5 | 38.6 |
| $SigLIP_{t+v}$ | 43.2 | 49.9 | 27.0 | 53.9 | 47.4 | 44.3 |

III but also all the other encoders from various VLMs. This superior performance establishes DinoV2 as a strong candidate for re-ranking, capable of substantially improving the quality of initially retrieved chunks and refining the overall retrieval process.

Table 13: We evaluate different encoders as visual re-rankers by using them instead of DinoV2 in our pipeline.

| Re-ranker | Birdsnap | CUB | iNaturalist | Indian Birds | NABirds | Average |
|---|---|---|---|---|---|---|
| | mRR@1/5/10 | mRR@1/5/10 | mRR@1/5/10 | mRR@1/5/10 | mRR@1/5/10 | mRR@1/5/10 |
| CLIP | 26.5/30.4/31.5 | 37.0/40.6/41.8 | 17.5/20.3/21.3 | 41.8/45.6/46.8 | 31.5/35.3/36.6 | 30.9/34.4/35.6 |
| OpenCLIP | 32.7/35.9/36.9 | 41.2/45.0/46.0 | 19.3/22.0/23.0 | 44.5/48.4/49.8 | 35.9/39.5/40.7 | 34.7/38.2/39.3 |
| SigLIP | 34.6/38.5/39.7 | 43.9/47.7/48.9 | 21.3/24.4/25.4 | 48.0/52.7/53.9 | 39.3/43.5/44.7 | 37.4/41.4/42.5 |
| Deit-III | 18.5/21.5/23.1 | 22.5/25.5/27.0 | 11.8/14.2/15.5 | 30.2/33.9/35.7 | 21.1/24.1/25.8 | 20.8/23.8/25.4 |
| DinoV2 | **48.9/52.6/53.8** | **58.0/62.3/63.1** | **34.8/38.7/39.8** | **68.1/72.9/73.6** | **52.0/56.6/57.6** | **52.3/56.6/57.6** |

## D    ABLATION WITH RETRIEVAL COMPONENTS.

In this section, we evaluate the contribution of each component in our retrieval pipeline within the VR-RAG framework using QWEN2.5-VL. Unlike table 6 in the main paper, here we analyze the performance drop when specific components are removed from the complete pipeline. As shown in table 15, each additional component, OpenCLIP, SigLIP, multimodal fusion, and the re-ranker, contributes to consistent improvements across all five benchmarks. The complete VR-RAG pipeline achieves the highest average performance, demonstrating that all components are integral to its effectiveness.

## E    WHY MULTI-MODAL RETRIEVAL WHEN RE-RANKING WITH DINOV2?

While the DinoV2 visual encoder exhibits strong unimodal performance( table 12), a critical question arises: can we bypass the multimodal retrieval step and rely solely on a cross-modal retrieval followed by DinoV2-based re-ranking? The effectiveness of this re-ranking hinges on the quality of the initial pool of top-ranked candidates. This approach should yield similar results to a full multimodal retrieval pipeline if the top-ranked candidates are already high-quality.

To investigate this, we evaluate VR-RAG without its multimodal representation, replacing the multimodal similarity score ($s^m$) with a cross-modal similarity ($s^c$). This is calculated by taking the dot product of the query image's visual features, $g(q)$, and the text chunk's features, $f(t)$, as shown in the equation below:

$$s^c = g(q)^\top f(t) \tag{3}$$

This initial step is followed by the re-ranking of the top r candidates in the same manner as the full pipeline(see eq. (2)). The results for all five avian datasets, the FishNet dataset, and the Pokémon dataset using Qwen 2.5VL are presented in Table 14.

The results show similar performance for both methods across all avian and Pokémon datasets. The average mRR@1 for these six datasets is 57.5% with cross-modal retrieval and 57.8% with multimodal retrieval. This similarity suggests that the initial cross-modal retrieval, in these cases, is robust enough to include high-quality candidates within the top 100 ranked species.

Table 14: Classification Results with cross-modal and multimodal representation followed by DinoV2 reranker.

| | Birdsnap | CUB | iNat | Ind Birds | NABirds | FishNet | Pokémon | Average |
|---|---|---|---|---|---|---|---|---|
| VR-RAG (cross-modal) | 53.8 | 59.9 | 36.7 | 68.5 | 57.5 | 10.5 | 86.8 | 53.4 |
| VR-RAG (multi-modal) | 51.9 | 60.3 | 35.5 | 72.2 | 56.3 | 18.7 | 86.3 | 54.5 |

Table 15: Ablation study: Ablation study on retrieval components in VR-RAG with QWEN2.5-VL. We progressively add each component and report the accuracy across five benchmarks. Each module contributes to performance gains, with the full pipeline achieving the best results.

| CLIP | OpenCLIP | SigLIP | Multimodal | Re-Ranker | Birdsnap | CUB | iNat | Ind. Birds | NABirds | Average |
|---|---|---|---|---|---|---|---|---|---|---|
| ✓ | ✗ | ✗ | ✗ | ✗ | 29.5 | 29.2 | 13.2 | 29.5 | 31.7 | 26.6 |
| ✓ | ✓ | ✗ | ✗ | ✗ | 38.9 | 44.3 | 18.9 | 36.2 | 43.5 | 36.4 |
| ✓ | ✓ | ✓ | ✗ | ✗ | 39.4 | 41.6 | 21.1 | 39.2 | 42.6 | 36.7 |
| ✓ | ✓ | ✓ | ✓ | ✗ | 46.0 | 53.3 | 26.9 | 58.7 | 51.3 | 47.2 |
| ✓ | ✓ | ✓ | ✓ | ✓ | **51.9** | **60.3** | **35.5** | **72.2** | **56.3** | **55.2** |

However, a significant performance gap emerges when the initial top-ranked candidates are of poor quality, as demonstrated by the FishNet dataset. In this case, the performance almost doubles in the multimodal setup. The initial cross-modal retrieval, which combines CLIP, OpenCLIP, and SigLip, yields a low mRR@1 of just 3.9%. Re-ranking this poor set with DinoV2 improves performance to 8.2% mRR@1.

In stark contrast, when we incorporate multimodal representations into the initial retrieval step, the mRR@1 score jumps to 14.5%. With the subsequent DinoV2 re-ranking, this performance further increases to an impressive 17.8% mRR@1, which is more than double the result of the unimodal approach. Furthermore, the inclusion of multimodal representation consistently improves results, even when using a single CLIP-like model, as detailed in Tables 12 and 15.

## F  RE-RANKING @ 100?

Figure 5 demonstrates that re-ranking significantly improves retrieval performance, but the benefits diminish rapidly after a certain point. As the number of candidates considered for re-ranking increases from 0 to 100, the average mRR@1, mRR@5, and mRR@10 all show a steep and substantial increase. For example, the mRR@1 score increases from a baseline of around 44% to over 51% with just 50 candidates, indicating that the re-ranking process effectively moves the correct answer to a higher position. However, the curves for all three metrics begin to plateau after approximately 100 candidates, indicating that the performance gains become marginal. Beyond this threshold, the added computational cost of re-ranking hundreds of additional items does not yield a meaningful increase in accuracy. This analysis confirms the conclusion that 100 candidates represent an optimal trade-off between efficiency and performance.

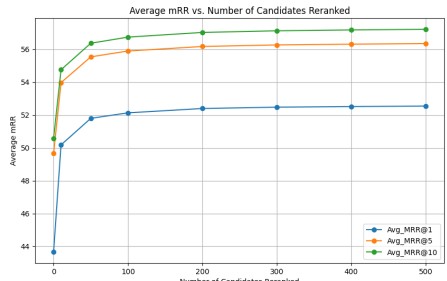

Figure 5: Effect of re-ranking on retrieval performance. Average mRR at ranks 1, 5, and 10 is shown as a function of the number of candidates re-ranked. Performance improves substantially when re-ranking the first few dozen candidates, but the gains plateau quickly. Beyond 100 candidates, the improvement is marginal, motivating the choice of 100 as the default setting in the main experiments.

## G  ANALYZING THE MIS-CLASSIFICATIONS.

In this section, we analyze the misclassifications to uncover interesting patterns. We leverage the taxonomic hierarchy by examining accuracy at the genus level. This approach reveals how frequently predictions fall within the same genus, as species within the same genus often share similar appearances. To calculate the genus accuracy, we map all the predicted species to the genus level and then calculate the accuracy. Instead of directly evaluating for genus, this approach allows us to analyze the behavior during species prediction. The results are presented in table 16. Our VR-RAG framework model significantly enhances genus-level accuracy across all LMMs, indicating that the retrieved information frequently belongs to the same genus. Consequently, the task of species classification for the LMM still remains chal-

Table 16: Classification Results at Genus Level: We map all the predicted species by different models to their genus. This helps us to understand if the misclassification happens because the task is difficult or if the models are making unexpected mistakes.

| | Birdsnap | CUB | iNat | Ind Birds | NABirds | Average |
|---|---|---|---|---|---|---|
| BioCLIP Stevens et al. (2024) | 43.4 | 46.2 | 33.6 | 29.2 | 21.5 | 32.0 |
| CLIP Radford et al. (2021) | 37.3 | 40.5 | 44.1 | 23.6 | 18.6 | 32.8 |
| OpenCLIP Ilharco et al. (2021) | 37.1 | 39.7 | 37.6 | 25.1 | 17.5 | 31.4 |
| SigLIP Zhai et al. (2023) | 40.6 | 38.6 | 44.4 | 28.7 | 27.7 | 36.0 |
| InternVL-3 Zhu et al. (2025) | 22.0 | 23.2 | 11.0 | 10.7 | 3.2 | 14.0 |
| MiniCPM-V-2.6 Yao et al. (2024) | 13.9 | 12.2 | 6.3 | 4.6 | 16.3 | 10.7 |
| Qwen2VL Yang et al. (2024) | 45.8 | 46.8 | 23.1 | 29.9 | 51.9 | 39.5 |
| Gemma-3n Yang et al. (2024) | 28.5 | 30.2 | 16.7 | 21.1 | 32.2 | 25.7 |
| Qwen2.5VL Qwen et al. (2025) | 47.0 | 48.4 | 42.2 | 25.8 | 8.1 | 34.3 |
| Qwen 2VL + VR-RAG (ours) | 73.7 | 77.6 | 58.6 | 86.3 | 78.4 | 74.9 |
| Qwen 2.5VL + VR-RAG (ours) | **74.8** | **79.9** | **60.3** | **86.6** | **79.2** | **76.2** |

lenging, as species within the same genus share numerous attributes. This inherent ambiguity may explain why refining the summary improves species recognition. By providing more discriminative

species descriptions, the model is better equipped to differentiate between species, even when they belong to the same genus.

As shown in fig. 19, we present two examples where the top-5 retrieved candidates are incorrect. The retrieved candidates exhibit notable similarities to the query image, highlighting the challenging, fine-grained nature of retrieving the correct species from a large pool of 11,202 candidates.

In fig. 20, we illustrate two instances where the correct species is successfully retrieved within the top 5 candidates. However, as is evident from the examples, the other four candidates also belong to the same genus. This high degree of similarity highlights the fine-grained nature of the retrieval task. The shared properties among these closely related species make it exceptionally challenging for the model to identify the correct candidate.

## H  TOP-K SELECTION.

In fig. 2, we showed the impact of increasing the number of top-k species candidates from $k = 5$ to $k = 15$ on the performance of the Qwen2.5-VL model. In fig. 6, we show the performance for the Gemma-3n model, and it shows a similar pattern as well. The average performance decreases from 47.8% at $k = 5$ to 47.0% at $k = 10$, and then to 46.3% at $k = 15$. And similar to Qwen2.5-VL, Gemma-3n also performs better at $k = 15$ with 47.0% rather than feeding in 5 Wikipedia articles of top-5 candidates, which results in an average of 33.6% accuracy. The drop is more severe for Gemma-3n as compared to Qwen2.5-Vl, highlighting the weakness of Gemma-3n struggling with bigger contexts.

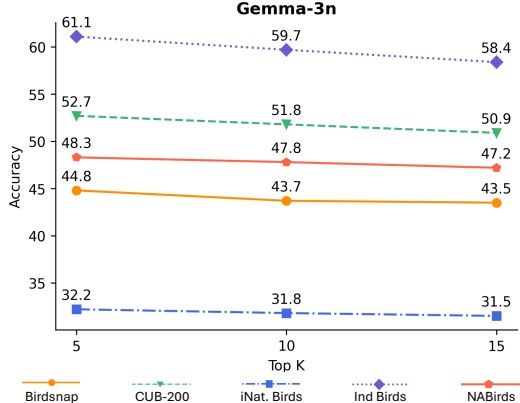

Figure 6: Top-k species selection ablation. We report the results for Gemma-3n on all five benchmarks by varying the value of k from 5 to 15.

## I  THE RE-RANKING MODULE

The re-ranking module begins by selecting the top-$r$ candidate species. For each of these species, the intra-modal similarity is calculated between the anchor images and the query image according to eq. (2). A visual illustration of this process is provided in fig. 7.

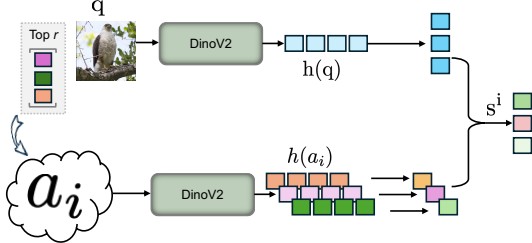

Figure 7: The visual representation of the re-ranker module.

## J  RE-RANKING ANALYSIS

We analyze two complementary scenarios:

- cases where re-ranking turns an initially correct retrieval into an incorrect one.
- cases where re-ranking successfully fixes an initially incorrect retrieval.

For both scenarios, we examine the score distributions in Figure 8, 9, 10, 11. We show the score distributions for the top 100 retrieval candidates before and after reranking, shown separately for reranking failure and success cases across five datasets. Birdsnap is not included due to the very low number of cases where re-ranking harms the initial retrieval.

Across all four datasets, we observe that the pre-reranking score distributions are strikingly similar for both failure and success cases: in every dataset, the distributions are mostly unimodal with a peak around 0.5. This indicates that the top-100 candidates have very similar retrieval scores before reranking, regardless of whether reranking eventually helps or harms the result.

In contrast, the post-reranking score distributions exhibit a clear divergence between failure and success cases:

- Failure cases: The distribution shifts to the right, with visually similar hard negatives receiving inflated similarity scores. This rightward shift implies that the reranker becomes overconfident in these distractors, elevating them above the correct candidate and causing it to drop out of the top-k.

- Success cases: The distribution instead shifts to the left, with a peak near 0.4 and a maximum extending to roughly 0.7. This leftward shift indicates that the reranker increases the separation between the correct candidate and the remaining ones. In these cases, the correct item is typically more visually distinct from hard negatives, allowing the reranker to impose a more discriminative ordering.

Across all datasets, CUB exhibits the widest score spread, suggesting that higher-scoring correct candidates are more distinct from lower-scoring incorrect candidates after reranking, which is followed by NABirds. Using this analysis, we derive a heuristic measure. We measure the difference between the maximum and median score of the candidates for both pre-re-ranking and post-re-ranking distributions and only apply re-ranking when $post - pre > \tau$, where $\tau$ is a threshold. Using this heuristic, the performance of CUB increases from 58.0 to 58.6 for mRRR@1, from 62.3 to 63.1 for mRR@5, and from 63.1 to 63.8 for mRR@10. Similarly, for NABirds, the scores changed from 52.0 to 52.4 for mRR@1, from 56.6 to 57.3 for mRR@5, and from 57.6 to 58.4 for mRR@10.

We visualize both scenarios in Figures 12 and 13, where we display the query image, the anchor image, and the top-5 candidates before and after re-ranking. Figure 12 illustrates a failure case where re-ranking harms the initial retrieval. As shown, all retrieved images are visually very similar, leaving little discriminative signal for the re-ranker to exploit.

In contrast, Figure 13 presents a successful case where re-ranking improves the initial results. Here, the pre–re-ranking candidates belong to the same genus but exhibit subtle visual differences. The re-ranker leverages these differences, allowing the correct candidate to receive a higher similarity score.

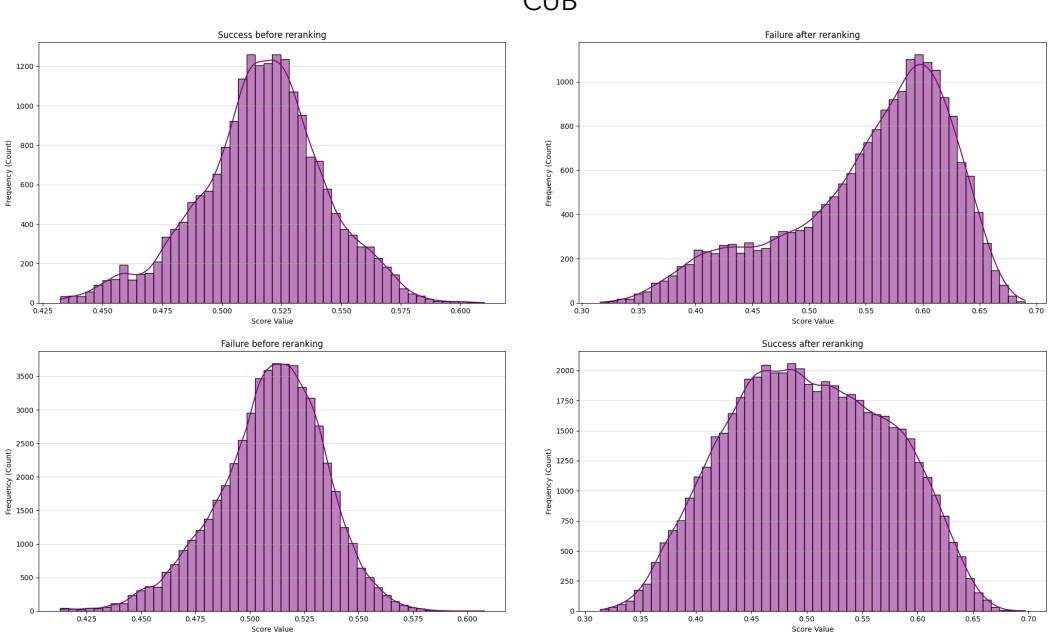

Figure 8: Similarity score distribution for CUB-200 dataset.

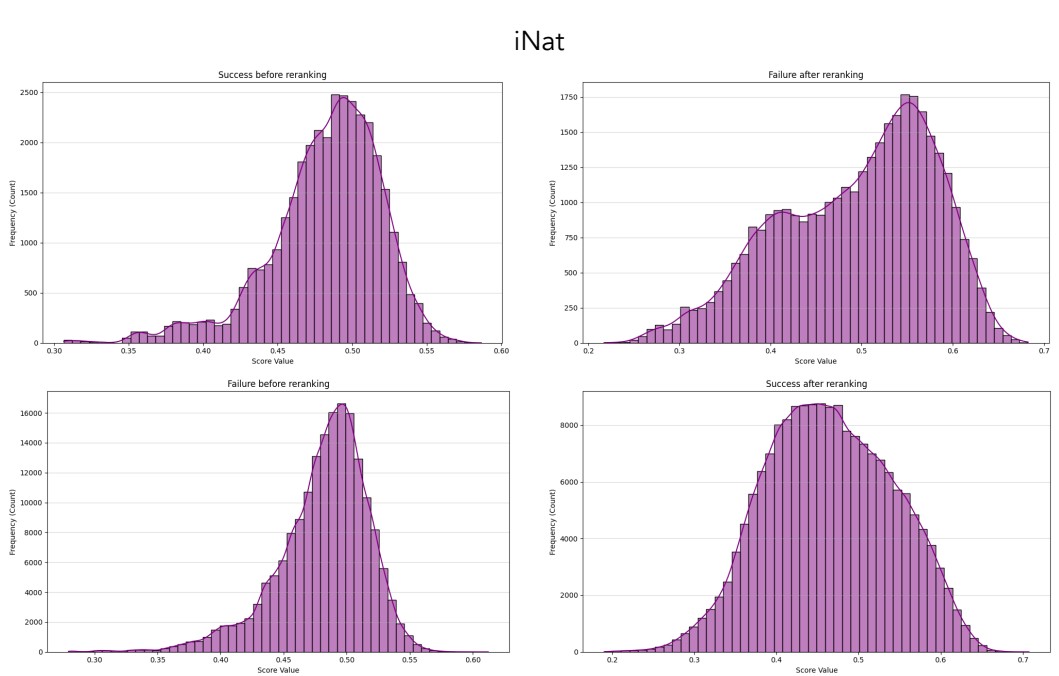

Figure 9: Similarity score distribution for iNaturalist dataset.

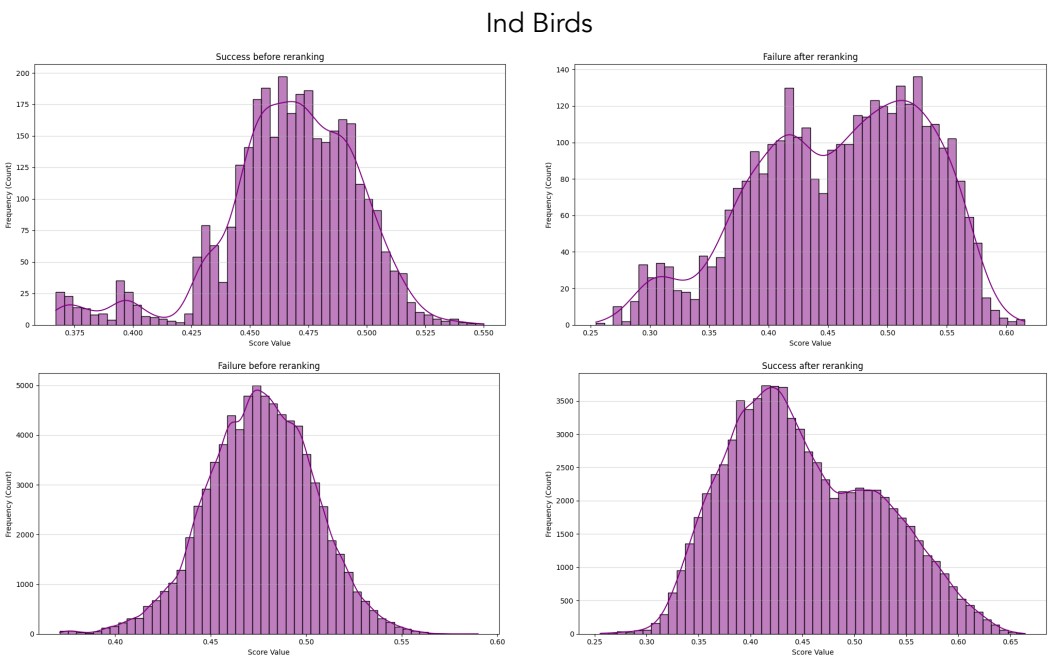

Figure 10: Similarity score distribution for Indian birds dataset.

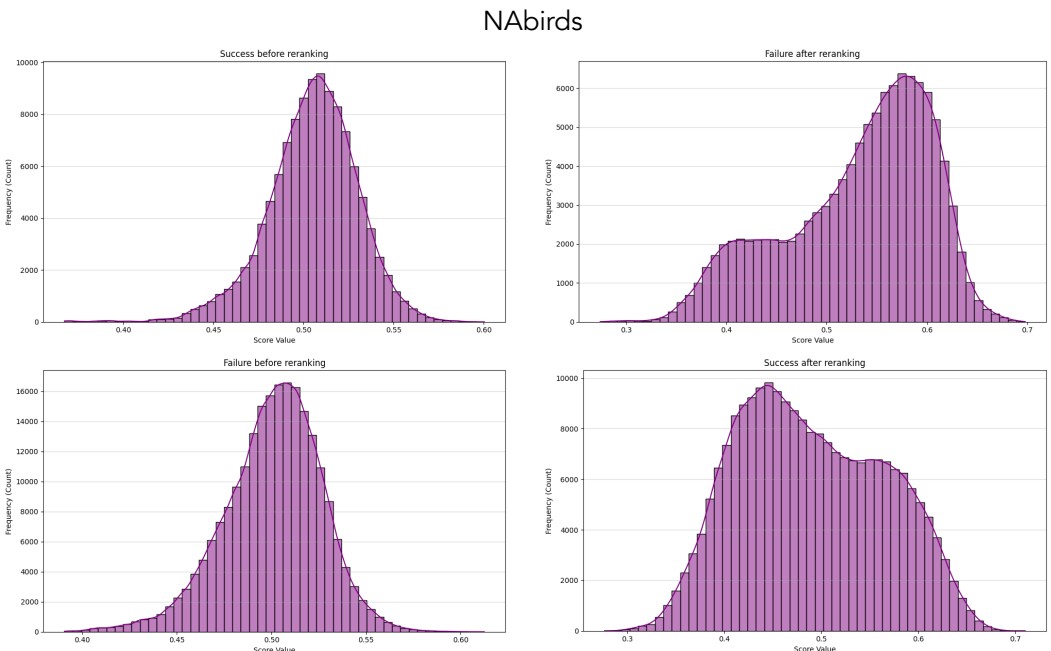

Figure 11: Similarity score distribution for NABirds dataset.

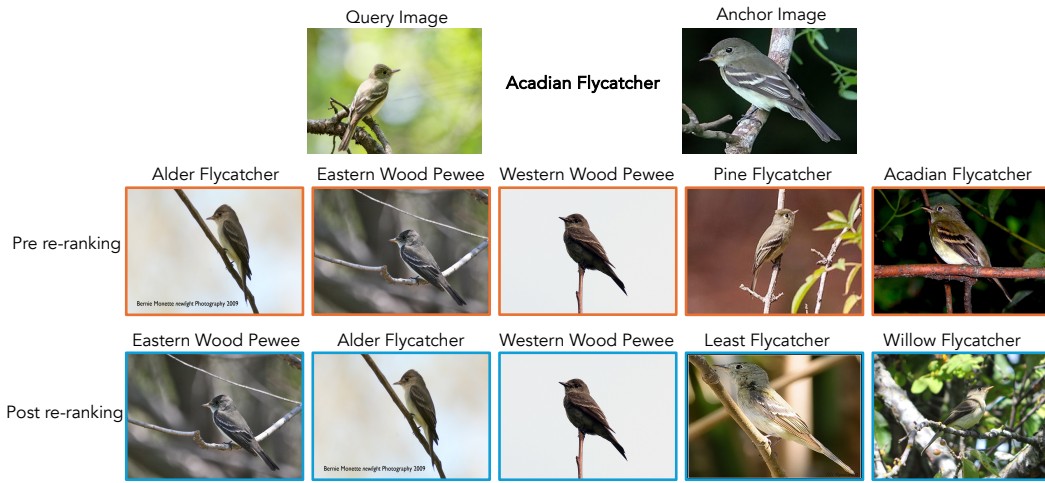

Figure 12: Visualization of a retrieval failure case where re-ranking harms the initial result. The Query Image (a bird) belongs to the Acadian Flycatcher class, as does the Anchor Image.

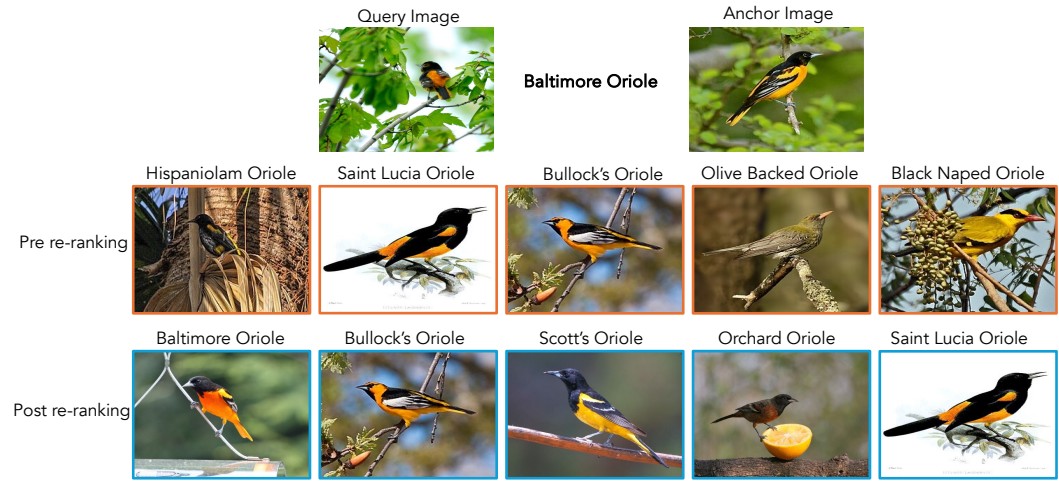

Figure 13: Visualization of a retrieval success case where re-ranking corrects the initial result. The Query Image (a bird) belongs to the Baltimore Oriole class, as does the Anchor Image.

## K  FUTURE WORK

While our work establishes a scalable framework for open-vocabulary species recognition, several directions remain open for further exploration.

**Expanding species coverage.**  Although birds benefit from near-complete Wikipedia documentation, many taxonomic groups, particularly marine organisms, insects, plants, and cryptic species lack high-quality public descriptions. Future efforts will focus on integrating alternative resources such as eBird, iNaturalist, GBIF, and field-guide literature to build a more comprehensive, multisource species knowledge base that extends beyond Wikipedia.

**Incorporating additional modalities.**  Our current pipeline focuses on images and text, but many species are better characterized through complementary modalities. Integrating audio (e.g., calls and songs), videos, and geolocation metadata could significantly improve recognition, especially for species that are visually similar but acoustically distinct.

**Scaling to richer tasks.**  We plan to extend VR-RAG beyond recognition and retrieval to more complex tasks such as fine-grained, large-scale visual question answering (VQA), multimodal reasoning, and open-world attribute prediction. These tasks would enable evaluating deeper understanding rather than surface-level matching.

**Intraspecies differentiation.**  Species often exhibit strong variation across sex, age, or seasonal plumage. A promising direction is to augment summaries with descriptors that explicitly distinguish males, females, juveniles, and breeding vs. non-breeding plumages. Modeling such intraspecific diversity remains a major challenge for current VLMs due to the lack of sufficient training data which is quite challenging to obtain.

**From single-object to multi-object scenes.**  Real-world observations frequently contain multiple individuals or multiple species interacting within the same frame. Extending VR-RAG to reason over multi-object, multi-species scenes—including detection, segmentation, and compositional reasoning would substantially improve its ecological applicability.

**Autonomous information acquisition.**  Finally, we aim to develop models that can identify when summaries lack critical information and automatically retrieve supplementary knowledge from the web or structured databases. This would shift VR-RAG from a static pipeline to an adaptive, self-improving system.

## L  HUMAN EVALUATION

### L.1  MATCHING DESCRIPTION WITH IMAGES

In this section, we analyze the discriminatory power of our generated species descriptions through a qualitative human study. Annotators of the forced-choice matching test also classified each of the 200 tasks as easy, medium, or hard based on the perceived time and effort required.

As illustrated in fig. 14, tasks classified as easy typically featured a description with a visual trait unique to the correct species, making the correct image immediately stand out. Tasks classified as medium, shown in fig. 15, required more focused attention, as the key distinguishing features were more subtle. For tasks classified as hard, shown in fig. 16, the distractors were extremely similar to the correct image, often differing by a single, minute attribute that required users to zoom in to discern the difference.

In fig. 17, we look at the cases where the annotators are inconclusive about their final decision, and in fig. 18, where annotators made incorrect choices. These scenarios occur under similar conditions, such as when a key visual trait was obscured in the image or when an annotator overlooked a critical detail in the description. For instance, in the first row of fig. 17, the annotator states that a clearer image of one of the confusing options would greatly solve the issue. And, in the second row of fig. 18, the annotator incorrectly matches a bird because they mistook the "pale crown stripe" as the white

line above the bird's eyes, forgetting that the pale strip is different as evident from the next sentence in description that tells us the pale stripe complements the white line and this is a feature present only in the correct image.

Ultimately, the difficulty of this fine-grained matching task shows the high quality and discriminatory power of our generated descriptions. They consistently include the specific, and often subtle, attributes necessary to differentiate species even in visually challenging scenarios.

The Bobolink (Dolichonyx oryzivorus) is a small New World blackbird distinguished by its unique plumage and size. Adult males are primarily black with striking creamy napes and white patches on their scapulars, lower backs, and rumps, making them easily recognizable. In contrast, adult females are mostly light brown with black streaks on their backs and flanks, and they have dark stripes on their heads, with darker wings and tails. Bobolinks measure between 5.9 to 8.3 inches (15–21 cm) in length, have a wingspan of about 10.6 inches (27 cm), and possess short, finch-like bills. These distinctive features, along with their unique coloration, help in identifying them from other similar bird species.

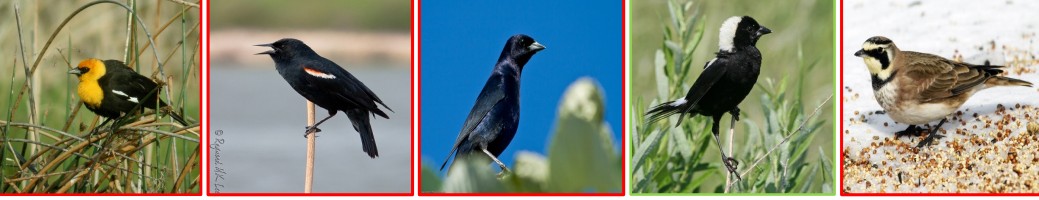

User: Primarily black removes last option, and only fourth one has a creamy nape, rest contradict. So I chose fourth.

The Geococcyx, commonly known as the roadrunner, is a distinctive bird found in the southwestern and south-central United States, Mexico, and Central America. It is a large, slender ground bird measuring 56 to 61 cm (22 to 24 in) in length, with a weight ranging from 230 to 430 g (8 to 15 oz). The roadrunner is characterized by its black-brown and white-streaked plumage, a prominent head crest, long legs, strong feet, and an oversized dark bill. Its tail is broad with white tips on the three outer feathers, and it has a unique bare patch of skin behind each eye, shaded blue in the front and red in the back. The roadrunner's zygodactyl feet leave distinct "X" track marks. Although capable of flight, it prefers to run, reaching speeds of up to 32 km/h (20 mph). In flight, its short, rounded wings reveal a white crescent in the primary feathers. These features, along with its unique vocalizations and habitat preferences, make the roadrunner easily identifiable.

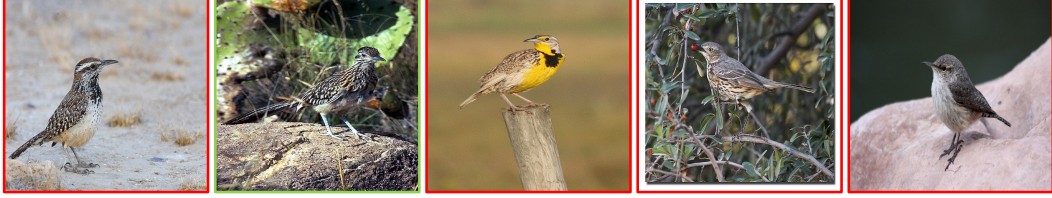

User: Black brown and white-streaked excludes third, long legs excludes fourth and tail with white tips excludes fifth, the x shape of foot excludes first and second has clear head crest as well. So, I select second.

Figure 14: Examples of description matching where the annotators classified the cases as easy. Wrong options are highlighted in red, and the correct option is highlighted in green.

The Barn Swallow (Hirundo rustica) is a small passerine bird easily identifiable by its distinctive physical features. It measures 17-19 cm in length, including its elongated outer tail feathers, and has a wingspan of 32-34.5 cm. The adult male showcases steel blue upperparts with a rufous forehead, chin, and throat, separated from its off-white underparts by a broad dark blue breast band. Its most striking feature is the deeply forked "swallow tail" adorned with white spots across the outer end. Females resemble males but have shorter tail streamers, less glossy blue upperparts, and paler underparts. Juveniles are browner with a paler rufous face and lack the long tail streamers. The Barn Swallow's combination of a red face and blue breast band distinguishes it from similar species, such as the African Hirundo species and the Welcome Swallow.

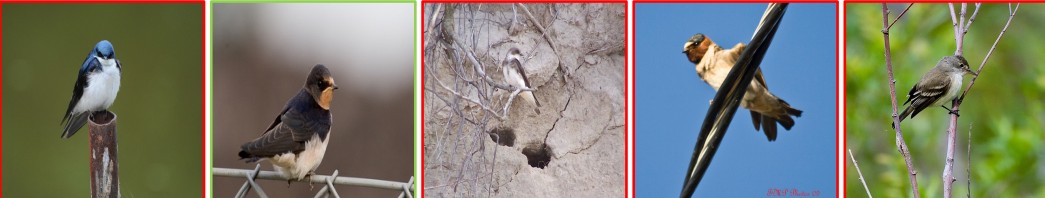

User: Steel blue is only true for first and second, so rest are removed. First is clearly blue but the second has also bluish color on its back, it also has rufous chin throat which first does not, so I select second.

The Bohemian Waxwing (Bombycilla garrulus) is a starling-sized bird, measuring 19-23 cm in length with a wingspan of 32-35.5 cm. It is characterized by its buff-grey plumage, a prominent pointed crest, and distinctive black face markings. The wings are notable for their striking patterns, featuring white and bright yellow with some feathers tipped in red, resembling sealing wax. The tail is short, ending in a bright yellow band bordered by black. The bird's beak is mainly black, and its legs are dark grey or black. Both males and females look similar, though females have slightly less pronounced markings. Juveniles are duller, with fewer red wing tips and a smaller black face mask. The Bohemian Waxwing can be distinguished from the smaller Cedar Waxwing by its size and more vivid wing patterns, and from the Japanese Waxwing by the absence of a red tail band and different wing markings. Its call is a high trill, and in flight, it resembles a common starling with its fast, direct movement.

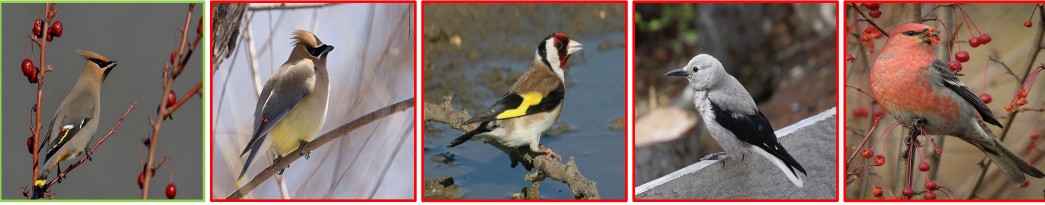

User: black face marking only present in first, second and third. The third option wing are only yellow and black and no white, so correct is either one or two. The second has no yellow on the wing, so first option should be correct.

Figure 15: Examples of description matching where the annotators classified the cases as medium. Wrong options are highlighted in red, and the correct option is highlighted in green.

The Common Tern (Sterna hirundo) is a medium-sized seabird distinguished by its slender build, long pointed wings, and deeply forked tail. Breeding adults exhibit pale grey upperparts and very pale grey underparts, with a striking black cap on the head. Their legs are orange-red, and the bill is narrow and pointed, varying in color from mostly red with a black tip to entirely black, depending on the subspecies. The upper wings are pale grey, developing a distinctive dark wedge as the summer progresses. In non-breeding adults, the forehead and underparts turn white, and the bill becomes all black or black with a red base. Juveniles have pale grey upper wings with a dark carpal bar and a ginger forehead that fades to white by autumn. The Common Tern can be differentiated from similar species like the Arctic Tern by its larger head, thicker neck, longer legs, and more triangular wings, as well as its more powerful flight. The Arctic Tern, in contrast, has greyer underparts and a consistently white wing throughout the summer. The Common Tern's tail does not extend beyond the wingtips when standing, unlike the Arctic and Roseate Terns. These features are crucial for identifying the Common Tern in photographs.

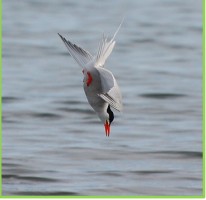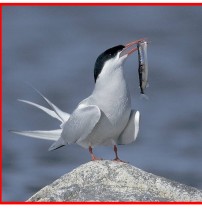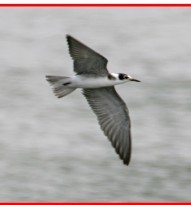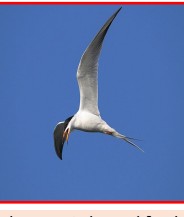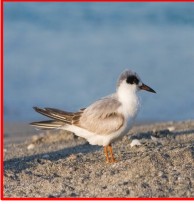

User: All are grey, but third has black legs so I remove it, second does not have black tip, forth has clear white underparts and not pale gray. It is either first or fifth, but given the color of fifth's beak and legs, first one is a much clearer match.

The Fish Crow (Corvus ossifragus) is a small crow species native to the eastern and southeastern United States, often found in wetland habitats. It is similar in appearance to the American Crow but is distinguishable by its smaller size, with a body length of 36-40 cm and a weight range of 247-320 grams. The Fish Crow has a silkier, smoother plumage with a blue or blue-green sheen on the upperparts and a greenish tint on the underparts. Its bill is somewhat slimmer, and it may have a small sharp hook at the end of the upper bill. When walking, Fish Crows appear to have shorter legs, and they tend to hunch and fluff their throat feathers when calling. The most distinctive feature is its call, a nasal "ark-ark-ark" or "nyuh unh," which contrasts with the American Crow's "caw caw." These vocal differences are often the most reliable way to differentiate between the two species.

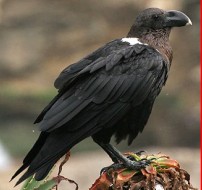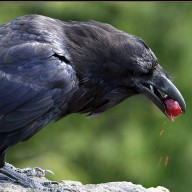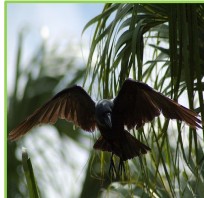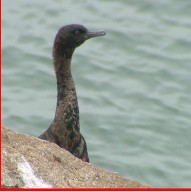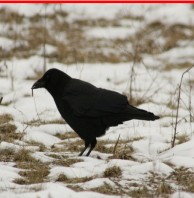

User: First and fifth do not show any bluish or greenish color, the fourth only shows brownish color but not any blue or green, so its either second or third. The bill of third seems slimmer but it is not clear, the legs of second seem larger than third and also has bigger beak, I feel the the colors in third image are not clear but all other images have some contradictions, so I select the third one.

Figure 16: Examples of description matching where the annotators classified the cases as hard. Wrong options are highlighted in red, and the correct option is highlighted in green.

The Common Raven (Corvus corax) is a large, all-black bird distinguished by its impressive size, with a length ranging from 54 to 71 cm (21 to 28 inches) and a wingspan of 116 to 153 cm (46 to 60 inches). It is one of the heaviest passerines, weighing between 0.69 to 2.25 kg (1.52 to 4.96 lbs). The raven's plumage is mostly iridescent black, with a dark brown iris and elongated, pointed throat feathers. Its large, slightly curved beak is one of the largest among passerines, and it has a long, wedge-shaped tail. In flight, ravens are recognized by their stable soaring style and the creaking sound of their feathers. They are often confused with crows but can be distinguished by their larger size, heavier beak, shaggy throat feathers, and distinct wedge-shaped tail. The raven's deep, resonant croak is also unique, setting it apart from the calls of other corvids.

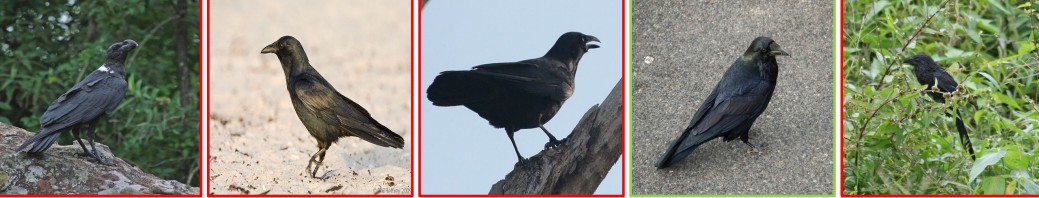

User: First one has white spot so its wrong, the third has no iridescent and no pointed throat feather, the second also lacks pointed throat feathers. The fifth is not clear to see if it has iridescent or not, so between fourth and fifth, they either have all features and no contradiction or low visibility. A closer image of the fifth one would help in selecting a single option.

The Cedar Waxwing (Bombycilla cedrorum) is a medium-sized bird, measuring about 6-7 inches in length with a wingspan of 8.7 to 11.8 inches. It is distinguished by its silky, shiny plumage in shades of brown, gray, and lemon-yellow, complemented by a subtle crest and a distinctive black mask bordered with white. A key identifying feature is the presence of small, red, wax-like droplets on the tips of its secondary flight feathers. The tail is typically yellow or orange, influenced by diet, and is short and square-tipped. The bird's beak is short and wide, and adults have a pale yellow belly. Immature birds are streaked and lack the adult's black mask. Cedar Waxwings are often seen in flocks, and their flight is strong and direct. They are known for their high-pitched whistles and buzzy trills.

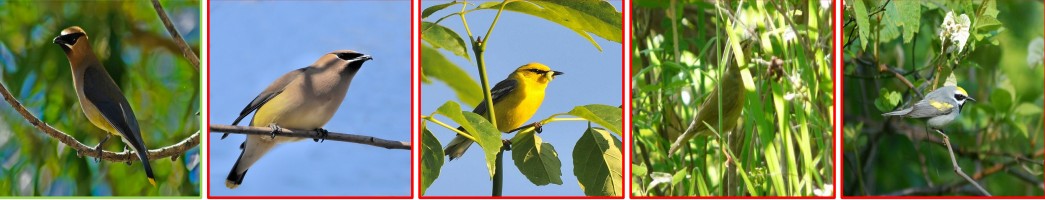

User: Third and fourth have a black line and not a mask, so it is one of first second or fifth. The red drop is not visible in either, probably its hidden in the image. Both first and second have a yellow part on the tail but fifth does not. The mask is bordered by white on first, in the second image, I am not sure if that color is white or is it the beak but there is also some white near the eyes. I am leaning more towards first but not sure.

Figure 17: Examples of description matching where the annotators are unable to pick one single option. Wrong options are highlighted in red, and the correct option is highlighted in green.

Frigatebirds, belonging to the family Fregatidae, are large seabirds found across tropical and subtropical oceans. They are characterized by their predominantly black plumage, long, deeply forked tails, and long, slender hooked bills. A distinctive feature of male frigatebirds is their red gular pouch, which they inflate during the breeding season to attract females. Females, on the other hand, have white underbellies. Frigatebirds have long, narrow wings that taper to points, with a wingspan that can reach up to 2.3 meters, giving them the largest wing area to body weight ratio of any bird. Their wings and tails create a distinctive 'W' silhouette in flight. These birds are known for their aerial prowess, rarely flapping their wings and instead soaring on wind currents. They do not settle on the ocean surface due to their minimal oil production, which would cause their feathers to become waterlogged. Frigatebirds are also known for their kleptoparasitic behavior, occasionally stealing food from other seabirds.

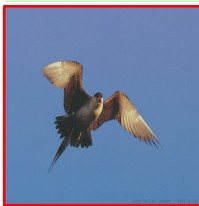 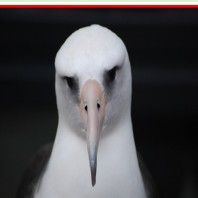 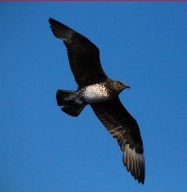 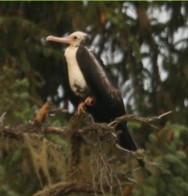 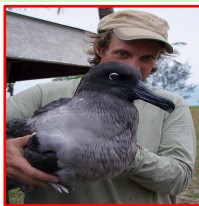

User: Since all options dont have the red pouch, the correct option should be a female. All images except second show black plumage, but the second image since it is not whole, it still can not be discarded. Fifth and first do not show white underbellies, so they can be discarded, even the fourth option has yellowish-white. The third option is in flight but not creating a W but an M, so I think the second one to be the correct.

The Clay-colored Sparrow (Spizella pallida) is a small sparrow native to North America, easily identifiable by its distinct plumage and markings. It features light brown upperparts with darker streaks on the back and pale underparts. A key characteristic is its pale crown stripe set against a dark brown crown, complemented by a white line above the eyes and a dark line through the eyes. The bird also has a light brown cheek patch and brown wings with noticeable wing bars. Its short bill is pale with a dark tip, and the back of its neck is gray. The Clay-colored Sparrow has a long tail, which adds to its distinctive silhouette. These features help distinguish it from similar species like the Chipping Sparrow and Brewer's Sparrow, especially outside the breeding season when they may flock together.

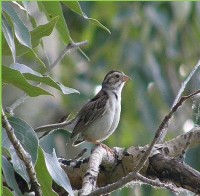 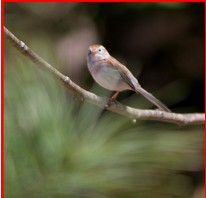 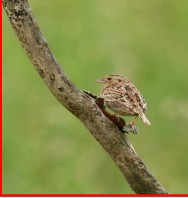 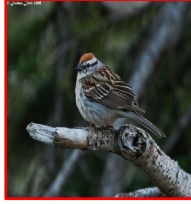 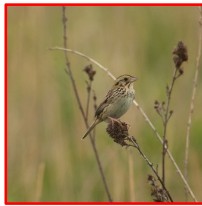

User: All the images show some form of brown upperparts with some darker streaks. The crown is not clearly visible in first, third and fifth option, but the fourth image has a brownish crown and a white line above the eyes also with a dark line passing through the eyes. So, I think the fourth option is the correct one.

Figure 18: Examples of description matching where the annotators pick the wrong option.

## L.2 RELIABILITY METHODOLOGY

The primary goal of our inter-annotator reliability analysis was to quantify the consistency of human judgments on the quality of the generated descriptions. To provide a direct and interpretable measure of consensus among the annotators, we use Quadratic Weighted Agreement (QWA), which is a direct application of the weighting principle from Cohen (1968), simplified by not correcting for chance agreement. The approach is described in algorithm 1.

---

**Algorithm 1** Calculation of Quadratic Weighted Agreement (QWA)

---

1: **Input:** A set of items $I$, where each item $i \in I$ has a list of scores $R_i = [s_1, s_2, ..., s_n]$ from $n$ annotators. The scoring scale has a minimum value $S_{min}$ and a maximum value $S_{max}$.
2: **Output:** The overall Quadratic Weighted Agreement score, $QWA_{overall}$.

3: Initialize an empty list, $A_{items}$, to store the agreement score for each item.
4: $D_{max} \leftarrow (S_{max} - S_{min})^2$           ▷ Calculate max squared difference once
5: **for all** item $i$ in $I$ **do**
6:      Let $R_i$ be the list of scores for item $i$.
7:      Initialize an empty list, $W_{pairs}$, to store weights for the current item.
8:      Let $P_i$ be the set of all unique pairs of scores from $R_i$.
9:      **for all** pair $(s_a, s_b)$ in $P_i$ **do**
10:          $D_{pair} \leftarrow (s_a - s_b)^2$
11:          $W_{pair} \leftarrow 1 - \frac{D_{pair}}{D_{max}}$          ▷ Quadratic weighting formula
12:          Add $W_{pair}$ to the list $W_{pairs}$.
13:      **if** $W_{pairs}$ is not empty **then**
14:          $A_{item} \leftarrow \text{average}(W_{pairs})$          ▷ Mean agreement for the item
15:          Add $A_{item}$ to the list $A_{items}$.
16: $QWA_{overall} \leftarrow \text{average}(A_{items})$          ▷ Grand mean across all items
17: **return** $QWA_{overall}$

---

# M QUALITATIVE DESCRIPTION SAMPLES

We show qualitative samples with images of species and their descriptions from our data. We show samples for various birds in fig. 21, and fig. 22. For our curated Pokémon dataset, we show examples in fig. 23.

**Acadian Flycatcher**

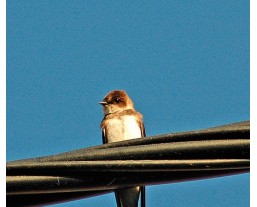

**Eastern Wood Pewee:** The Eastern Wood Pewee (Contopus virens) is a small North American tyrant flycatcher, measuring 13.5–15 cm in length with a wingspan of 23–26 cm. It is characterized by its gray-olive upperparts and light underparts with an olive wash on the breast. Notable features include two pale wing bars and long primary remiges, giving the wings a slim, pointed appearance. The upper part of the bill is dark, while the lower part is yellowish. This bird is often confused with the Western Wood Pewee, but can be distinguished by its distinct mournful whistled "pee-a'wee" call. Unlike the Eastern Phoebe, which lacks defined wing bars and frequently bobs its tail, the Eastern Wood Pewee has a more pronounced primary projection. The Least Flycatcher, another similar species, has a bold eye ring and shorter primary projection, appearing blunt-winged.

**Alder Flycatcher:** The Alder Flycatcher (Empidonax alnorum) is a small bird, measuring 13–17 cm in length with a wingspan of 8.3–9.4 cm. It is characterized by its dull greenish-olive upperparts and a slightly darker olive-green crown. The throat is white, contrasted by a darker breast band, and it features thin white eye rings. Its bill is distinctive, with a pinkish or yellow-orange lower mandible and a black upper mandible. The wings are black with white wing bars and white-edged tertials. Juveniles display brownish upperparts and yellow underparts with yellowish-brown or buff wingbars. The Alder Flycatcher is similar to the Eastern Wood-Pewee and Eastern Phoebe but can be distinguished by its smaller size and unique upward tail flicking behavior. Vocalizations, particularly its "fee-bee-o" song, are crucial for identification, as visual differentiation from the nearly identical Willow Flycatcher can be challenging.

**Least Flycatcher:** The Least Flycatcher (Empidonax minimus) is a small bird, measuring 12 to 14 cm in height with a wingspan of 19 to 22 cm, and weighing around 10.3 grams. It is the smallest of the Empidonax flycatchers in eastern North America. Its plumage is characterized by a dull olive-gray back and a whitish belly, which is notably brighter than other Empidonax species. Distinctive features include two white wing bars and white eye rings. The beak is short, with a yellowish lower mandible. Juveniles resemble adults but have slightly darker wing bars with tawny or olive hues. While these physical traits are shared with other flycatchers, the Least Flycatcher is best identified by its distinctive "che-bec" call and its preferred habitats.

**Galapagos flycatcher:** The Galápagos flycatcher, or large-billed flycatcher (Myiarchus magnirostris), is a small bird endemic to the Galápagos Islands, measuring 15 to 16 cm in length and weighing between 12 to 18.5 grams. It is the smallest member of its genus. This bird is distinguished by its relatively large bill, which is a key feature for identification. The plumage is generally a muted brown with lighter underparts, and it has a subtle crest on its head. The wings are rounded, typical of flycatchers, aiding in its agile flight. The Galápagos flycatcher is known for its curious behavior, often approaching humans and interacting with camera lenses, mistaking its reflection for another bird.

**Western Wood Pewee:** The Western Wood Pewee (Contopus sordidulus) is a small tyrant flycatcher, measuring 5.5–6.3 inches in length with a wingspan of about 10.2 inches. It is characterized by its gray-olive upperparts and light underparts, with an olive wash on the breast. Distinctive features include two wing bars and a dark bill with yellow at the base of the lower mandible. This bird closely resembles the Eastern Wood Pewee, but can be distinguished by its call, a loud buzzy "peeer," and its song, which consists of three rapid descending "tsees" ending with a descending "peeer." These features are crucial for visual identification, especially when observing the bird in its natural habitat.

**Bank Swallow**

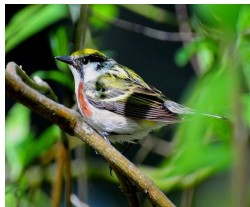

**Northern Rough Winged Swallow:** The Northern Rough-winged Swallow (Stelgidopteryx serripennis) is a small swallow, measuring 13–15 cm in length with a wingspan of 27–30 cm. It is characterized by its brown upperparts and white underparts, with a distinctive brownish-grey wash on the throat. The swallow has a small bill and a forked tail. A key feature is the serrated edge feathers on its wings, which are not visible unless the bird is held. Males have longer and broader undertail coverts compared to females, and their outer primary wings have hooked barbs. Juveniles can be identified by their reddish-brown wing-bars. This species is similar to the Bank Swallow but can be distinguished by its dusky throat and breast. The Northern Rough-winged Swallow is often found near water, where it feeds on flying insects.

**Streak-throated swallow:** The streak-throated swallow, or Indian cliff swallow (Petrochelidon fluvicola), is a small passerine bird approximately 11 cm (4.3 in) in length, notable for its distinctive physical features. It has a dull chestnut forehead and crown, contrasting with its steel-blue upperparts. The underparts are whitish, and the bird is characterized by brown streaks on the throat and chest, which are key identifiers. Its tail is square-ended, and the rump is pale brown. The streak-throated swallow is often seen in large flocks, especially on cold mornings, perched closely on wires. Its vocalizations include a twittering chirp when perched and a sharp "trr-trr" in flight. These features, along with its habitat preference for open areas near water bodies, help distinguish it from other swallows.

**Brown Chested Martin:** The brown-chested martin (Progne tapera) is a medium-sized swallow distinguished by its brown chest and upperparts, contrasting with a paler underbelly. It features a streamlined body typical of swallows, with long, pointed wings that are often seen bowed during flight. The bird's tail is slightly forked, displaying white on the sides, which is particularly noticeable when it swoops at low heights. Its beak is short and broad, adapted for catching insects mid-flight. This martin is often observed in large flocks, especially around dusk, and is known for nesting in burrows dug into banks or using old nests of other birds like the hornero. These characteristics, along with its habitat preferences, help distinguish it from other similar species.

**Brown-bellied swallow:** The Brown-bellied Swallow (Orochelidon murina) is a small bird distinguished by its unique plumage and body shape, which aids in its identification. This species features a distinctive brown belly, contrasting with its darker upperparts, which are typically a shade of blue or black. The swallow's wings are long and pointed, characteristic of the Hirundinidae family, allowing for agile flight. Its beak is short and slightly curved, adapted for catching insects mid-air. The bird's tail is slightly forked, a common trait among swallows, but less pronounced than in some other species. These features, combined with its habitat preference for high-elevation shrublands and grasslands in countries like Bolivia, Colombia, Ecuador, Peru, and Venezuela, make the Brown-bellied Swallow identifiable in its natural environment.

**Chilean swallow:** The Chilean swallow (Tachycineta leucopyga) is a small bird, approximately 13 centimeters (5.1 inches) in length, with a weight ranging from 15 to 20 grams (0.53 to 0.71 ounces). It is characterized by its glossy blue-black upperparts and white underparts, with a distinctive white rump. The wings and tail are black, featuring white tips on the inner secondaries and tertials, while the underwing coverts and auxiliaries are grey. Both sexes appear similar, though juveniles are duller and browner. Unlike the similar white-rumped swallow, the Chilean swallow lacks a white forehead and has bluer upperparts. Its bill and legs are black, aiding in its identification.

**Chestnut Sided Warbler**

**Reiser's tyrannulet:** Reiser's tyrannulet (Phyllomyias reiseri) is a small bird measuring 11 to 11.5 cm in length and weighing 7 to 8 grams. It features a bright olive crown, nape, back, and rump with faint grayish tips on the crown feathers. The bird's face is marked by yellowish white lores, supercilium, and cheeks, with a thin olive line running through the eye. Its wings are dusky with pale yellowish edges on the flight feathers and the ends of the coverts, forming two distinct bars on the closed wing. The tail is dusky olive, while the throat and lower face are whitish, and the underparts are pale yellow with faint olive streaks on the breast and sides. The short, rounded bill has a blackish maxilla and a black-tipped pinkish to white mandible, and the legs and feet are gray. These features, along with its pale brown iris, help distinguish Reiser's tyrannulet from other similar species.

**Yellow Throated Vireo:** The Yellow-throated Vireo (Vireo flavifrons) is a small songbird distinguished by its striking coloration and features. It measures between 5.1 to 5.9 inches in length with a wingspan of about 9.1 inches. This bird is primarily olive on the head and upperparts, contrasted by a vivid yellow throat and white belly. One of its most distinctive features is the yellow "spectacles" around its dark eyes. The tail and wings are dark, accented with white wing bars, and it possesses thick blue-grey legs and a stout bill. These characteristics, particularly the yellow throat and spectacles, make it easily identifiable among similar species.

**Pine Warbler:** The Pine Warbler (Setophaga pinus) is a small songbird distinguished by its white belly, two white wing bars, and dark legs. It features a thin, relatively long pointed bill and yellowish "spectacles" around its eyes. Adult males are notable for their olive upperparts and bright yellow throats and breasts, while females and immatures have olive-brown upperparts with paler throats and breasts. This bird's appearance can be confused with the yellow-throated vireo due to similar coloration. The Pine Warbler's song is a musical trill, and its calls are slurred chips. These birds are typically found in open pine woods and are known for their slow foraging on tree trunks and branches, often poking their bills into pine cones.

**White Eyed Vireo:** The White-eyed Vireo (Vireo griseus) is a small songbird, measuring 4.3–5.1 inches in length with a wingspan of 6.7 inches. It is distinguished by its greyish olive head and back, white underparts with yellow flanks, and dark wings and tail. Notably, it features two white wing bars on each wing and striking white irises surrounded by yellow spectacles, which are key identifiers. Both sexes appear similar. This bird's compact size, distinctive eye markings, and color pattern make it recognizable among other vireos.

**Providencia vireo:** The Providencia vireo (Vireo approximans) is a distinctive bird distinguished by its robust, thick bill, which is a key feature for identification. It has a predominantly olive-green plumage on its upperparts, with a contrasting pale yellow or whitish underbelly. The wings are short and rounded, typical of vireos, and often display faint wing bars. Its head is marked by a subtle, yet distinctive, white or pale eyebrow stripe above the eye, which helps differentiate it from similar species. The combination of its thick bill, olive-green and yellowish coloration, and the subtle facial markings are essential for identifying the Providencia vireo, especially in its natural habitats of subtropical or tropical moist forests on Isla de Providencia.

Figure 19: Examples of top-5 retrieved candidates that do not contain the correct species description. The retrieved images and their corresponding text descriptions show significant visual and textual similarities to the query image, highlighting the fine-grained and challenging nature of the retrieval task.

**Bewick Wren**

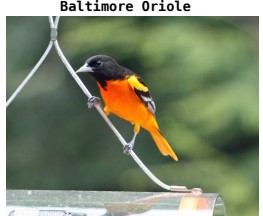

**Marsh Wren:** The Marsh Wren (Cistothorus palustris) is a small songbird native to North America, easily identifiable by its distinctive physical features. It measures between 3.9 to 5.5 inches in length with a wingspan of approximately 5.9 inches. This bird has brown upperparts, a light brown belly and flanks, and a white throat and breast. Its back is marked with black and white stripes, and it sports a dark cap with a prominent white line above the eyes. The Marsh Wren has a short, thin bill, which is a key feature distinguishing it from similar species. Its habitat typically includes marshes with tall vegetation like cattails, where it is often seen foraging close to water.

**Bewick Wren:** The Bewick's Wren (Thryomanes bewickii) is a small North American bird, measuring about 14 cm (5.5 inches) in length with a wingspan of 18 cm. It is characterized by its grey-brown upperparts and white underparts, with a distinctive long white eyebrow. Its long tail, which is frequently moved and features black bars with white corners, is a key identifying feature. The bird has a long, slender, slightly curved beak. Juveniles resemble adults but have shorter, stockier beaks and may show faint speckling on the underbelly. Both males and females look similar. The Bewick's Wren is often confused with the Carolina Wren but can be distinguished by its longer tail and the white tips. It is a vocal species, with males known for their varied and melodious songs.

**Carolina Wren:** The Carolina Wren (Thryothorus ludovicianus) is a relatively large wren, measuring 12.5 to 14 cm in length with a wingspan of about 29 cm. It is characterized by its rich brown plumage, which appears more chestnut on the rump and upper tail coverts, and a distinctive white superciliary streak bordered by thin black lines. The bird's underparts are a warm buff color, transitioning from gray on the chin and throat. Its wings and tail feature darker brown barring, and the tail has 18 to 20 bars. The Carolina Wren has a slightly curved, lemon-colored upper mandible, and flesh-colored legs. Males are generally larger than females, with longer wing chords. The species can be confused with Bewick's Wren, which is smaller with a longer tail and grayer-brown above. The Carolina Wren's energetic behavior, distinctive song, and preference for dense cover in forests and suburban areas further aid in its identification.

**Saint Lucia wren:** The Saint Lucia wren (Troglodytes mesoleucus) is a very small bird native to the Caribbean island of Saint Lucia. It is part of the wren family, Troglodytidae, and is distinguished by its diminutive size and tendency to disappear into crevices, a behavior linked to its name meaning "hole dweller." This species features a plumage with a distinctive "middling-white" or streaked white pattern, which sets it apart from other wrens. The Saint Lucia wren has a compact body, a short tail, and a slender, slightly curved beak, typical of wrens. Its wings are short and rounded, aiding in its agile movements through dense vegetation. These physical characteristics, along with its unique vocalizations and behaviors, help in identifying the Saint Lucia wren from photographs.

**Kalinago wren:** The Kalinago wren (Troglodytes martinicensis) is a very small bird native to the Caribbean island of Dominica, previously also found on Martinique and Guadeloupe. This wren is distinguished by its compact size and characteristic behavior of disappearing into crevices. It features a short, slightly curved beak ideal for insect hunting. The plumage is generally brown with subtle variations, providing excellent camouflage against its natural habitat. The wings are short and rounded, typical of wrens, aiding in quick, agile movements through dense foliage. The Kalinago wren's distinct vocalizations and genetic differences set it apart from the closely related northern house wren.

**Baltimore Oriole**

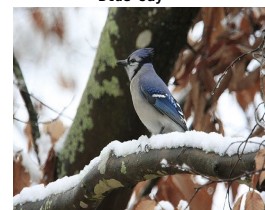

**Baltimore Oriole:** The Baltimore Oriole (Icterus galbula) is a medium-sized passerine bird, measuring 17–22 cm in length with a wingspan of 23–32 cm. It is distinguished by its striking plumage, particularly in males, which feature vibrant orange underparts, shoulder patches, and rump, contrasted with black on the rest of the body. Females and juveniles are more subdued, with yellow-brown upper parts and dull orange-yellow on the breast and belly. Both sexes have distinctive white wing bars. The oriole's thick, pointed bill is typical of the icterid family. This bird is often found high in deciduous trees and is known for its sweet, whistling song. The Baltimore Oriole's preference for ripe, dark-colored fruits and its unique feeding behavior, such as "gaping" into soft fruits, further aid in its identification.

**Bullocks Oriole:** Bullock's Oriole (Icterus bullockii) is a small, sexually dimorphic bird distinguished by its striking plumage and size. Adult males are notable for their vibrant orange and black coloration, featuring a black throat patch, a black line extending from each eye to the crown, and a white wing bar. Their underparts, breast, and face are orange or yellow, contrasting with the black back, wings, and tail. The tail is mostly black with the outermost rectrices tipped in orange, forming a "T" shape. Females, in contrast, have gray-brown upperparts, duller yellow on the breast and underparts, and an olive crown, lacking the black eye-line seen in males. Both sexes have a pointed bill with a straight culmen, and the wingspan measures about 12.2 inches. Juveniles resemble adult females but have darker wings and a pink or whitish bill. These orioles are often found in riparian corridors and open woodlands, and they are known for their distinctive song, which is similar to but faster and harsher than that of the Baltimore Oriole.

**Scott Oriole:** The Scott's Oriole (Icterus parisorum) is a medium-sized bird distinguished by its striking plumage and distinct physical features. It measures approximately 9.1 inches (23 cm) in length, with a wingspan of 12.6 inches (32 cm), and weighs between 1.1 to 1.4 ounces (32–41 g). This oriole is notable for its vibrant yellow and black coloration; the males typically exhibit a bright yellow body with contrasting black head, throat, and upper back, while females and juveniles display a more subdued yellow with olive tones. The beak is slender and slightly curved, typical of the oriole family, aiding in its identification. The wings are pointed, enhancing its agile flight. These distinctive features make the Scott's Oriole easily recognizable, especially in its preferred habitats in the Southwestern United States and parts of Mexico.

**Orange oriole:** The Orange Oriole (Icterus auratus) is a small bird, measuring about 19–21 cm (7.5–8.3 in) in length, distinguished by its vibrant orange plumage and black markings. The males feature a striking black region between their eyes and bill, with a black stripe across the upper breast, while the head and body are predominantly orange. The wings display a unique pattern with orange lesser feathers and white medium and greater layers. Females are generally duller, with an orange-yellow wash and green mantles. The bird has a slender body, long wings, and a pointed beak, which it uses to pry into fruits. The Orange Oriole's habitat in the Yucatán Peninsula influences its plumage color due to dietary variations. This species is non-migratory, residing in forests, woodlands, and abandoned farmlands.

**Altamira Oriole:** The Altamira oriole (Icterus gularis) is the largest oriole in its genus, measuring about 25 cm (9.8 in) in length. Both males and females share similar striking features, including a black mandible, throat, back, and long tail. Their wings are predominantly black with white-fringed flight feathers, forming a distinctive single white wing bar and white wing spots when folded. The secondary coverts display bright orange epaulets, while the underside is a vibrant orange or yellowish-orange. Immature birds have an olive back and a dull yellow head and body, with first-year birds resembling adults but with an olive back and yellow-olive tail. These features, along with their large size and vivid coloration, make the Altamira oriole easily distinguishable from other orioles.

**Blue Jay**

**Blue Jay:** The Blue Jay (Cyanocitta cristata) is a striking bird native to eastern North America, easily identifiable by its vibrant plumage and distinctive features. Measuring 22–30 cm in length with a wingspan of 34–43 cm, it showcases a prominent blue crest on its head, which can be raised or lowered depending on its mood. The bird's plumage is a captivating lavender-blue to mid-blue on the crest, back, wings, and tail, contrasted by a white face and underparts. A notable black, U-shaped collar encircles its neck, extending to the sides of the head. The wings and tail are intricately barred with black, sky-blue, and white patterns. Its bill, legs, and eyes are black, and both sexes appear similar, with males being slightly larger. The Blue Jay's coloration is due to structural coloration rather than pigments, as the blue hue results from light interference within the feather structure. This bird is known for its noisy and bold behavior, often mimicking the calls of hawks and other sounds.

**Eurasian Jay:** The Eurasian jay (Garrulus glandarius) is a medium-sized bird in the crow family, measuring 34–35 cm in length with a wingspan of 52–58 cm. It is distinguished by its pinkish-brown body plumage, a whitish throat bordered by black moustache stripes, and a striking bright blue patch on the upper wing with fine black bars. The bird's tail is predominantly black, and its forehead and crown feature whitish feathers with black streaks. The Eurasian jay's complex wing coloration includes black and white bars, and its rump is white. This bird is known for its harsh, rasping screech and its ability to mimic other species' calls. It inhabits mixed woodlands, particularly those with oaks, and is a habitual acorn hoarder. The Eurasian jay's distinct racial forms vary across its vast range, with notable differences in crown streaking, mantle color, and facial markings.

**Black-headed Jay:** The Black-headed Jay (Garrulus lanceolatus) is a slender bird similar in size to the Eurasian Jay but distinguished by its shorter, thicker bill and more pronounced crest. Its most notable feature is the black top of its head, which contrasts with its otherwise lighter plumage. The bird also has a longer tail compared to its relatives. It inhabits regions from eastern Afghanistan across the Himalayas to Nepal and Bhutan, favoring wooded areas with open spaces. The Black-headed Jay's vocalizations are akin to the Eurasian Jay's, characterized by loud screeches with longer pauses. These features, along with its habitat preferences, make it identifiable from photographs.

**White-throated Magpie-Jay:** The White-throated Magpie-Jay (Cyanocorax formosus) is a strikingly large bird, measuring between 43 and 56 cm in length, with a distinctive long tail and a slightly curved crest of feathers on its head. The crest is black in the nominate subspecies, with blue or white margins in others. It features a white face contrasted by a black crown and a narrow black band around the throat, with a small black drop below the eye. The breast, belly, and underside of the rump are white, while the wings, mantle, and tail are a vibrant blue, with whitish margins on the tail. The bird's legs and eyes are black, and it has a grey bill. Females resemble males but have duller plumage on top, a narrower chest band, and a shorter tail. These features, along with its gregarious nature and noisy behavior, make the White-throated Magpie-Jay easily identifiable.

**Steller's Jay:** Steller's jay (Cyanocitta stelleri) is a striking bird native to western North America, easily identifiable by its unique physical characteristics. It measures about 30–34 cm (12–13 in) in length and weighs between 100–40 g (3.5–4.9 oz). This species is the only crested jay west of the Rocky Mountains, featuring a prominent crest that is more pronounced in northern populations. The head varies in color from blackish-brown to dark blue, depending on the subspecies, with lighter streaks on the forehead. The body transitions from a dark head to a silvery blue on the shoulders and lower breast, while the primaries and tail are a rich blue with darker barring. Steller's jays in the eastern part of their range may have white markings on the head, particularly over the eyes, whereas those further west have light blue markers or none at all. The bird's slender bill and longer legs, along with its vibrant plumage and distinctive vocalizations, make it distinguishable from the closely related blue jay.

Figure 20: Examples of top-5 retrieved candidates that contain the correct species description ranked. The other four candidates belong to the same genus as the query image and exhibit significant visual and textual similarities to it, highlighting the fine-grained and challenging nature of the retrieval task.

 Abbott's babbler (Malacocincla abbotti) is a small, nondescript bird measuring 12–13 cm in length, characterized by its short tail and heavy bill. Its plumage is primarily drab olive-brown, with a grayish-white throat and breast, and a white belly center. The flanks and undertail coverts are a distinctive rusty color, while the supercilium and lores are pale gray. Juveniles have darker rufescent-brown crowns and upperparts. Both sexes appear similar. This species is often found in low vegetation near streams, favoring areas with tree ferns and tangled undergrowth. Its distinctive calls, which include a series of rich, fluty notes, aid in identification. The subspecies M. a. krishnarajui is noted for its darker russet tail and rump compared to the nominate subspecies. Abbott's babbler is typically seen foraging in pairs close to the ground.

 The Abyssinian owl, also known as the African long-eared owl (Asio abyssinicus), is a medium-sized owl distinguished by its dark brown eyes and black bill. It features gray eyebrows and prominent dark brown ear-tufts with white edges, which are slightly centrally located on its head. This owl is darker in appearance compared to the long-eared owl (Asio otus), with which it does not share a range. The Abyssinian owl's strong claws allow it to prey on a variety of animals, including smaller birds, field mice, and shrews. It inhabits open grasslands or moorlands with oak or cedar forests, typically found in mountain valleys and gorges at elevations up to 3,900 meters.

 The Afep pigeon (Columba unicincta), also known as the African wood-pigeon or gray wood-pigeon, is a medium-sized bird measuring 35 to 36 cm in length and weighing between 356 and 490 grams. It is characterized by its gray neck and body, with darker gray wings and tail, and a distinctive white throat and belly. The breast displays a buff-pink hue. Notably, the Afep pigeon has red eyes and orbital rings, which are key features for identification. Its call is loud and distinctive, often described as "doo doo doo" or "whu whu whu whu-WHU." These pigeons are native to the African tropical rainforest regions flanking the Dahomey Gap.

 The African blue flycatcher, also known as the blue-crested flycatcher (Elminia longicauda), is a small, dainty bird distinguished by its bright blue plumage and long tail. It features a short crest and its upper parts, including the tail, are a vibrant blue that transitions between blue and cyan. The bird's lores and flight feathers are black, with the latter edged in blue, while its underparts are a greyish blue that fades to whitish on the belly. The bill and legs are black, adding to its distinctive appearance. Juveniles are duller, with faint greyish spotting on the head and wing coverts. Measuring 15–18 cm in length and weighing 7–12 g, this flycatcher is often seen foraging gracefully with a fanned tail and half-open wings. These features, along with its unique coloration, make it easily identifiable among other bird species.

 Allen's hummingbird (Selasphorus sasin) is a small, vibrant bird measuring 3 to 3.5 inches in length and weighing 2 to 4 grams. The male is distinguished by its green back and forehead, with rust-colored (rufous) flanks, rump, and tail, and an iridescent orange-red throat. In contrast, females and immature birds are mostly green with rufous only on the tail, which has white tips, and lack the iridescent throat patch, instead having speckled throats. A key distinguishing feature of the adult male Allen's hummingbird is the absence of a notch in the second rectrix, which helps differentiate it from the similar rufous hummingbird. Allen's hummingbirds are known for their energetic courtship displays and aggressive territorial behavior. They are primarily found along the coastal regions from California to Oregon, with some populations migrating to central Mexico for the winter.

 The Alor boobook (Ninox plesseni) is a small owl native to the Pantar and Alor Islands in the eastern Lesser Sunda Islands. It is distinguished by its compact size and rounded head, typical of the boobook owls. The plumage is predominantly brown with intricate patterns of lighter and darker streaks, providing excellent camouflage against tree bark. Its facial disc is less pronounced compared to other owls, with subtle white markings around the eyes. The eyes are large and dark, set against a relatively small, hooked beak. The wings are rounded, aiding in silent flight, and the tail is short. These features, along with its unique vocalizations, help differentiate the Alor boobook from other similar species.

 The Chinese egret (Egretta eulophotes) is a medium-sized bird, averaging 68 cm in height, with distinctive all-white plumage throughout its life, similar to the little egret. Key identifying features include a dusky bill base and yellow-green lores and legs outside the breeding season, while the iris remains yellow. During the breeding season, the Chinese egret develops a striking luxuriant crest over 11 cm long, along with long lanceolate breast plumes and dorsal aigrettes extending beyond the tail. The bill turns a bright orange-yellow, the lores become bright blue, and the legs turn black with yellow feet. These seasonal changes, particularly the vibrant breeding plumage and unique crest, help distinguish it from other egret species.

 The Coleto (Sarcops calvus) is a distinctive bird species belonging to the family Sturnidae. It is characterized by its medium size and unique physical features that make it easily identifiable. One of the most striking characteristics of the Coleto is its bare, pinkish head, which contrasts sharply with its glossy black plumage. The bird's beak is short and stout, adapted for its varied diet. Its wings are rounded, aiding in agile flight through its forested habitat. The Coleto's overall appearance, with its combination of a bald head and sleek black body, sets it apart from other similar bird species, making it a unique subject for birdwatchers and photographers alike.

 The collared lory (Vini solitaria) is a vibrant parrot species endemic to Fiji, easily identifiable by its striking plumage. Measuring approximately 20 cm (7.9 in) in length, this bird features bright red underparts and face, a dark purple crown, and greenish upperparts. The nape is lime green with some elongated red feathers, and the lower abdomen is purple. Its bill is yellow-orange, feet are pink-orange, and irises are orange-red. Males and females are similar, though females have a paler crown with a greenish hue towards the back. Juveniles are duller with vague purple striations on the upper abdomen and breast, and they have a brown beak and pale brown irises. The collared lory's adaptation to urban environments, such as Suva, and its fast, straight flight with quick shallow wingbeats further distinguish it.

 The collared sunbird (Hedydipna collaris) is a tiny bird, measuring only 9\u201310 cm (3.5\u20133.9 in) in length, and is part of the Nectariniidae family. It is characterized by its short, thin, down-curved bill and brush-tipped tubular tongue, adaptations for nectar feeding. The adult male is distinguished by its glossy green upperparts and head, a yellow belly, and a narrow purple breast band. In contrast, the female is a duller green above and entirely yellow below. These birds have a fast and direct flight on short wings and are typically found in forests near water across sub-Saharan Africa. The collared sunbird's unique combination of size, bill shape, and vibrant plumage makes it identifiable from other similar species.

 The Comoro black parrot (Coracopsis sibilans) is a medium-sized parrot distinguished by its predominantly dark plumage, which appears blackish with subtle brownish tones. This bird features a robust, slightly curved beak that is well-suited for its diet. Its wings are relatively broad and rounded, aiding in its flight through the dense forests of the Comoros, where it is endemic. The parrot's tail is moderately long and squared off at the end. Unlike many other parrots, the Comoro black parrot lacks the bright, vivid colors often associated with the family, making its dark, muted coloration a key identifying feature.

 The collared finchbill (Spizixos semitorques) is a distinctive songbird known for its unique physical characteristics. It features a striking black head and a contrasting white collar around its neck, which is a key identifier. The bird's body is predominantly olive-green, providing a vibrant contrast to its darker head. It has a stout, finch-like beak that is well-suited for its frugivorous diet. The wings are rounded, and the tail is relatively short, aiding in its agile flight through forested hills. This species is typically found in China, Taiwan, Japan, and Vietnam, and is often seen in moderate elevation forests. The collared finchbill's combination of a black head, white collar, and olive-green body makes it easily distinguishable from other similar species.

 The Beautiful Fruit Dove (Ptilinopus pulchellus), also known as the rose-fronted pigeon or crimson-capped fruit dove, is a small bird measuring approximately 19 cm (7.5 inches) in length. It is primarily green with distinctive features that aid in its identification: a striking red crown, a whitish throat, a greenish-yellow bill, and purplish-red feet. Its breast is blue-grey, transitioning to a yellowish-orange belly, with a notable reddish-purple patch in between. Both male and female birds share these vibrant color patterns. This species is native to the rainforests of New Guinea and nearby islands in West Papua, Indonesia. The Beautiful Fruit Dove's unique combination of colors and small size make it distinguishable from other fruit doves.

 The common scimitarbill (Rhinopomastus cyanomelas) is a distinctive bird characterized by its dark blue plumage, which can appear almost black in certain lighting. A key feature for identification is its long, curved beak, which is black in adults and grey in juveniles. This bird's sleek, elongated body and scimitar-shaped bill set it apart from other species. It is typically found across various regions in Africa, including Angola, Botswana, and South Africa, among others. When identifying this bird from a photograph, look for its unique combination of dark blue coloration and the pronounced curve of its beak.

 The Corsican finch (Carduelis corsicana) is a small bird in the true finch family, Fringillidae, distinguished by its unique plumage and coloration. It features dark-streaked brown upperparts, which contrast with its brighter yellow underparts, a key characteristic that sets it apart from the similar citril finch. The Corsican finch has a typical finch-like beak, suited for seed eating, and its overall size is small, consistent with other finches. This bird is endemic to the Mediterranean islands, including Corsica, Sardinia, Elba, Capraia, and Gorgona, which can also aid in its identification.

 The cream-eyed bulbul (Pycnonotus pseudosimplex) is a distinctive bird endemic to Borneo, notable for its unique cream-colored eyes, which set it apart from other bulbuls. This species is generally found in old-growth hill forests, emphasizing its habitat specificity. It shares a close genetic relationship with the ashy-fronted bulbul of Palawan, rather than the red-eyed Bornean population of the cream-vented bulbul. The cream-eyed bulbul's plumage and overall appearance are similar to other bulbuls, but the eye color is a key distinguishing feature for identification.

Figure 21: Examples of bird species with their descriptions.

The Alpine chough, or yellow-billed chough (Pyrrhocorax graculus), is a distinctive bird in the crow family, easily identified by its glossy black plumage, bright yellow beak, and red legs. It measures 37–39 cm in length with a wingspan of 75–85 cm, featuring a proportionally longer tail and shorter wings compared to its close relative, the red-billed chough. The Alpine chough's flight is buoyant and acrobatic, characterized by loose, deep wing beats and high maneuverability, often seen soaring in updrafts near cliffs. Its calls are unique, with rippling "preep" and whistled "sweeeooo" sounds, differing from the crow-like calls of similar species. Juveniles are duller, with a less vibrant yellow bill and brownish legs. This bird is unlikely to be confused with others due to its combination of size, coloration, and vocalizations.

The Andaman bulbul (Brachypodius fuscoflavescens) is a distinctive bird endemic to the Andaman Islands, easily identifiable by its primarily olive-yellow plumage. A key feature is its olive-colored head, which sets it apart from other bulbuls. This bird is medium-sized, typical of the bulbul family, and has a slender, slightly curved beak suited for its diet of small fruits, berries, and insects. The combination of its unique coloration and feeding habits makes the Andaman bulbul a standout species for bird enthusiasts and photographers alike.

The Andean lapwing (Vanellus resplendens) is a distinctive bird approximately 33 cm (13 in) in length, with a weight ranging from 193 to 230 g (6.8 to 8.1 oz). It features a creamy gray head and neck, accented by a dark brownish gray patch around the eye. Its upperparts are a striking bronzy green with a purple patch on the wing coverts, while the breast is dark gray and the belly is white. The bird's bill is pinkish orange with a black tip, and both its eyes and legs are reddish. Juveniles differ slightly, with a brownish head and neck, buff mottling on the breast, and pale buff fringes on the upperparts feathers. These characteristics, along with its unique coloration and size, make the Andean lapwing easily identifiable among similar species.

The ashy tit (Melaniparus cinerascens) is a small bird distinguished by its predominantly ashy-grey plumage, which gives it its name. It features a distinctive black cap and bib, contrasting sharply with its lighter grey body. The wings and tail are darker grey, providing a subtle contrast to the rest of its plumage. Its beak is short and stout, typical of the Paridae family, adapted for its diet. The ashy tit's size is relatively small, similar to other tits, making it agile in its natural habitats of subtropical or tropical dry forests and savannas. These key characteristics, along with its geographical range in southern Africa, help distinguish it from other similar species.

The Australian masked owl (Tyto novaehollandiae) is a distinctive barn owl species found in Southern New Guinea and non-desert areas of Australia. It is characterized by a striking white, heart-shaped facial mask surrounded by brown feathers. The dorsal plumage is predominantly brown with light gray spots on the upper back, while the front is white with brown spots. The owl exhibits sexual dimorphism, with females being larger and darker than males. Males typically weigh between 420 to 800 grams and measure 330 to 410 mm in length, whereas females weigh between 545 to 1,260 grams and measure 390 to 500 mm. The wingspan can reach up to 1,280 mm, particularly in southern female masked owls. The eye color ranges from black to dark brown. The Tasmanian subspecies is notably the largest within the barn owl family. These owls are nocturnal and primarily hunt terrestrial prey, although they can also capture prey from trees or in flight.

The Australian zebra finch (Taeniopygia castanotis) is a small, striking bird native to Central Australia, easily recognizable by its distinctive physical features. Adult males are particularly notable for their bright orange cheek patches, red beaks, and bold black and white patterns, including a black tear drop stripe below the eye and a black band across the chest. Females, while similar in size, have more subdued coloring with orange beaks and lack the vibrant cheek patches. Both sexes have a grey body with a white belly, and their wings display a pattern of black and white bars. The zebra finch's beak is short and conical, ideal for seed eating. These birds are small, typically measuring around 10–11 cm in length, with a wingspan of approximately 12–14 cm. Their compact size, combined with their distinctive plumage and vocalizations, makes them easily identifiable in their natural arid habitats or in captivity.

The Bahia antwren (Herpsilochmus pileatus) is a small bird, measuring 10.5 to 11 cm in length and weighing around 9 grams. It is distinguished by its striking plumage and distinct markings. Adult males feature a black crown and nape, with a prominent white to pale gray supercilium and a black streak through the eye. Their upperparts are gray with white-edged blackish scapulars and a notable white patch between them. The wings are black with white-tipped coverts and white-edged flight feathers, while the tail is black with white tips and edges on the outermost feathers. The underparts are predominantly white with a gray wash. In contrast, adult females have a buffish forehead and a black and white streaked crown, with gray upperparts tinged with olive and underparts that have an ochraceous hue. These distinctive features, along with their habitat in coastal Bahia, Brazil, make the Bahia antwren identifiable from other similar species.

Baird's Sparrow (Centronyx bairdii) is a small, brown-streaked sparrow native to North America, easily identifiable by its distinctive broad ochre central crown stripe. The bird's face is a yellow-brown color with subtle black markings, and it features a narrow band of brown streaks on its chest. Unlike the LeConte's Sparrow, Baird's Sparrow is larger and lacks orange facial coloration. It also differs from Henslow's Sparrow by not having green facial tones, and from the Savannah Sparrow by having less heavy streaking and lacking an extra white marking on the head. Adult Baird's Sparrows are about 12 cm (5 inches) in length, with a wingspan of approximately 23 cm (9 inches), and weigh between 17–21 grams. These characteristics make it distinguishable from other sparrows in its habitat.

The Bali myna (Leucopsar rothschildi) is a medium-sized, stocky bird measuring up to 25 cm (9.8 in) in length. It is distinguished by its almost entirely white plumage, featuring a long, drooping crest and black tips on its wings and tail. A notable characteristic is the blue bare skin surrounding its eyes, complemented by greyish legs and a yellow bill. Both male and female Bali mynas appear similar, although the male typically has a longer crest. This bird is critically endangered, with fewer than 50 adults estimated to exist in the wild as of 2020. The Bali myna's unique combination of white plumage, blue eye skin, and black wing and tail tips make it easily identifiable among other starling species.

The Baltimore Oriole (Icterus galbula) is a medium-sized passerine bird, measuring 17–22 cm in length with a wingspan of 23–32 cm. It is distinguished by its striking plumage, particularly in males, which feature vibrant orange underparts, shoulder patches, and rump, contrasted with black on the rest of the body. Females and juveniles are more subdued, with yellow-brown upper parts and dull orange-yellow on the breast and belly. Both sexes have distinctive white wing bars. The oriole's thick, pointed bill is typical of the icterid family. This bird is often found high in deciduous trees and is known for its sweet, whistling song. The Baltimore Oriole's preference for ripe, dark-colored fruits and its unique feeding behavior, such as "gaping" into soft fruits, further aid in its identification.

The band-tailed fruiteater (Pipreola intermedia) is a distinctive bird found in the montane forests of Bolivia and Peru. It measures about 19 cm (7 in) in length and is characterized by its plump green body with black, chevron-shaped markings on the flanks. A key feature is its tail, which has a green base, a prominent black band, and a whitish tip. The adult male is particularly notable for its glossy black head and bib, bordered by a bright yellow collar, while the female lacks these features. Both sexes have a green iris and red beak and legs. The underparts are yellowish with green mottling or streaks. This species can be distinguished from the similar green-and-black fruiteater by its larger size and more distinct yellow collar.

Barau's petrel (Pterodroma baraui) is a medium-sized seabird, approximately 40 cm in length, notable for its striking plumage and distinct markings. It features a predominantly white underside and forehead, contrasting with its dark upper parts. A key identifying characteristic is the moderately distinct "M" pattern across its wings and back, which is typical of many gadfly petrels. The bird's bill is black, adding to its distinctive appearance. Barau's petrel is primarily found in the Indian Ocean, with its main breeding site on the island of Réunion. These features, along with its unique breeding habits and restricted range, help distinguish it from other similar seabird species.

The collared grosbeak (Mycerobas affinis) is a large finch, measuring 22 to 24 cm in length, with a robust body and a strong, thick bill measuring 2.7 to 2.9 cm. Adult males are distinctive with their glossy black head, upper wings, and tail, contrasted by an earthy brown collar and rich, deep yellow plumage on the rest of the body. Females, on the other hand, are olive-green on the back and yellowish below, lacking the black facial markings. Juveniles resemble the adult females. The wingspan ranges from 12.1 to 13.8 cm, and the tail measures 8.7 to 9.7 cm. This species is often found in mountainous deciduous or mixed forests across the Himalayas and surrounding regions. The collared grosbeak's flight is characterized by a fast, direct, and sometimes undulating style, and it is known for its mellow, rapid flight call and sharp alarm call.

Bell's vireo (Vireo bellii) is a small songbird measuring 4.5–4.9 inches in length with a wingspan of 6.7–7.5 inches. It is characterized by its dull olive-gray upperparts and whitish underparts, which provide a subtle contrast. A faint white eye ring and faint wing bars are distinguishing features that can aid in identification. The bird has a relatively small and slender beak typical of vireos, suited for its insectivorous diet. Bell's vireo is often found in dense shrubbery, which it uses for nesting, and its plumage provides effective camouflage in such environments. These features, along with its size and coloration, help differentiate it from other similar bird species.

The Blue-faced Honeyeater (Entomyzon cyanotis), also known as the Bananabird, is a large honeyeater, measuring around 29.5 cm (11.6 in) in length. It is easily identifiable by its striking blue facial skin, which is bare and contrasts with its predominantly black head and throat. The bird's plumage features olive upperparts and white underparts, with a distinctive white stripe around the nape and cheeks. Its wings are broad with rounded tips, and it has a medium squarish tail. The bill is sturdy, slightly downcurved, and shorter than the skull, measuring 3 to 3.5 cm (1.2 to 1.4 in) in length. Juveniles can be distinguished by their yellow or green facial patches, which transition to the adult blue as they mature. The Blue-faced Honeyeater's unique coloration and size set it apart from other honeyeaters, making it a distinctive species for visual identification.

The chestnut-headed nunlet (Nonnula amaurocephala) is a small bird, measuring 14 to 15 cm in length and weighing 15 to 16 grams. It is distinguished by its bright rufous head, upper mantle, breast, and upper belly, which contrast with its plain dull brown back, wings, and tail. The rump has a subtle olive wash, and the rufous upper belly transitions to a whitish lower belly. Notable features include a mostly black bill, red eyes, and lead gray feet. This bird is endemic to a specific region in Brazil and is typically found in the understory of seasonally flooded igapó forests. Its unique coloration and habitat preference make it identifiable among similar species.

Figure 22: Examples of more bird species with their descriptions.

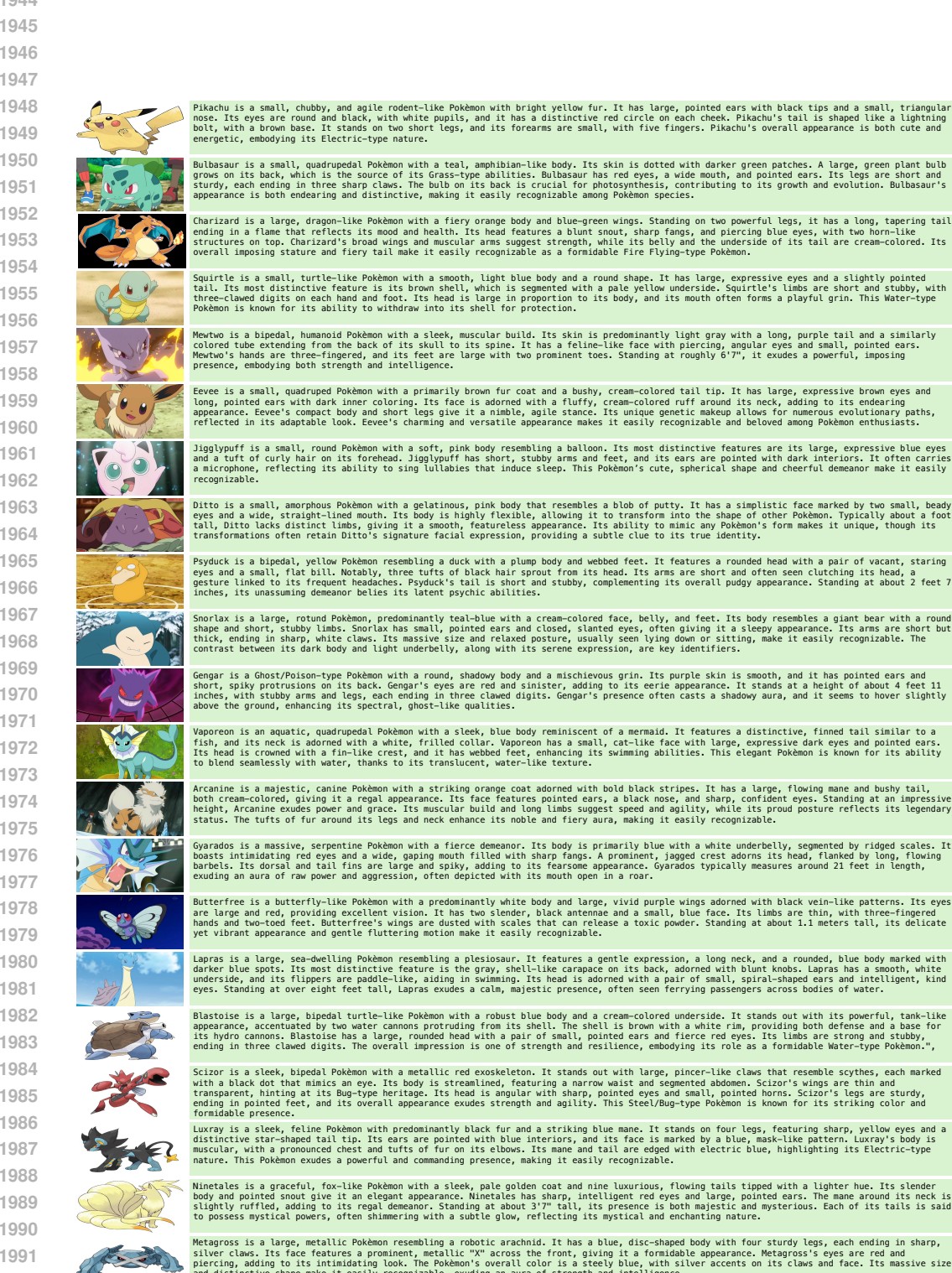

Figure 23: Pokémon species with their descriptions.

