# OpenReview forum: "Neural Catalog: Scaling Species Recognition with Catalog of Life–Augmented Generation"
_ICLR.cc/2026/Conference — Submitted to ICLR 2026_

### Official Review · Reviewer_2xRi · 2025-10-29

**Soundness:** 2
**Presentation:** 3
**Contribution:** 2
**Rating:** 4
**Confidence:** 3

**Summary:**

This paper introduces NeuralCatalog, a retrieval-augmented generation (RAG) framework for open-vocabulary bird species recognition. It leverages Wikipedia as an external knowledge source, retrieving concise species descriptions to complement visual features from images. By integrating retrieval, re-ranking, and multimodal reasoning within a large language model, NeuralCatalog enables accurate recognition of both seen and unseen species, achieving state-of-the-art performance in bird species recognition.

**Strengths:**

1. The problem of open-vocabulary species recognition is practical and of scientific interest.
2. The improvements obtained are strong and non-trivial in nature.

**Weaknesses:**

I believe this paper has several weaknesses that are not adequately addressed in the current submission. My main concerns center on positioning and evaluation.

W1. While the paper’s motivation around biological species recognition is strong, the inclusion of a new Pokémon dataset feels abrupt and insufficiently justified. As presented, it appears more like an attempt to expand the technical scope rather than a well-motivated extension of the main problem.

W2. The evaluation against encoder-only models raises concerns about fairness. Since the proposed method ensembles multiple encoder-based VLMs (e.g., CLIP), comparing against these same models individually in Tables 2 and 3 seems misaligned. Moreover, the paper omits several relevant baselines from recent zero-shot prompting and retrieval works [1,2,3], which weakens the empirical positioning of the method.

W3. The retrieval pool includes over 11,000 species—substantially larger than prior benchmarks—yet it remains unclear whether species appearing in test sets (e.g., CUB, iNat) are excluded from the retrieval corpus during evaluation. If not, this introduces a fairness issue, as the model could indirectly access prior knowledge of test species during inference.

References
[1] Prompting Scientific Names for Zero-Shot Species Recognition - EMNLP 2023

[2] Visual Classification via Description from Large Language Models - ICLR 2023

[3] Learning Customized Visual Models with Retrieval-Augmented Knowledge - CVPR 2023

**Questions:**

Please refer to the weaknesses outlined above.

---

> ### Author Response · Authors · 2025-11-20
> **Rebuttal by Authors Part [1/2]**
>
> We appreciate your time and feedback. Below, we have carefully addressed your specific concerns and questions. We have updated the main paper to reflect the changes.
> ___
>
> > **W1. While the paper’s motivation around biological species recognition is strong, the inclusion of a new Pokémon dataset feels abrupt and insufficiently justified. As presented, it appears more like an attempt to expand the technical scope rather than a well-motivated extension of the main problem.**
>
> We understand the concern: introducing a Pokémon dataset may seem tangential to the core problem of recognizing biological species. However, we incorporated it to demonstrate the Domain-Agnosticism of VR-RAG. A central claim of our work is that VR-RAG is not specific to natural species but is a generalizable framework for open-vocabulary recognition across domains. By demonstrating performance not only on birds (11,202 species) and marine species (FishNet), but also on a synthetic/virtual domain (Pokémon), we illustrate that our pipeline is effective even when taxonomy is fictional or non-biological. This cross-domain evaluation strengthens the methodological contribution, as it demonstrates that the framework does not rely on biological signals, but rather on visual discriminativeness and textual descriptions.
>
> > **W2. The evaluation against encoder-only models raises concerns about fairness. Since the proposed method ensembles multiple encoder-based VLMs (e.g., CLIP), comparing against these same models individually in Tables 2 and 3 seems misaligned. Moreover, the paper omits several relevant baselines from recent zero-shot prompting and retrieval works [1,2,3], which weakens the empirical positioning of the method.**
>
> We thank the reviewer for bringing these relevant works to our attention. We have added a new section, Section 6.4, where we compare to these baselines. Below, we summarize how they relate to our setting and how VR-RAG interacts with them.
>
> Both [1] and [2] propose training-free improvements to CLIP:
>
> * [1] Prompting Scientific Names (EMNLP 2023): Their main finding is that common names outperform scientific names in zero-shot CLIP classification. Since all our datasets already use common names, their improvement is equivalent to using CLIP with our retrieval setup. This baseline is included in our ablation (Table 15 in the appendix).
>
> * [2] Description-based Classification (ICLR 2023): This work uses GPT-generated descriptions for recognition. We differ in that we utilize real-world Wikipedia-derived descriptions, along with visual re-ranking. We evaluate it directly and also apply our visual re-ranking on top of their method. VR-RAG yields large gains, demonstrating that our approach is complementary and can enhance prior prompting-based techniques.
>
> * [3] Customized Visual Models with RAK (CVPR 2023) improves CLIP through training-based retrieval augmentation. However, we observe that in our open-vocabulary setting, its standalone performance is limited. After integrating VR-RAG, its performance improves substantially. This again shows that VR-RAG is compatible with and boosts existing retrieval-augmented models.
>
> VR-RAG is not positioned as a competitor to [1,2,3] but as a general enhancement module that can be applied on top of them. Across all three cases, VR-RAG yields consistent and significant improvements, strengthening the empirical positioning of our method.
>
> | Method | Average Accuracy |
> |-----|------|
> | Menon et. al. | 30.6 |
> | Menon et. al. + Visual Reranking| 49.4 |
> | Liu et. al. | 26.6 |
> | Liu et. al. + Visual Reranking| 44.1 |
>
>
> We acknowledge that comparing VR-RAG, which integrates multiple encoders, directly against individual encoder models may appear imbalanced. However, we demonstrate that even with a single encoder, such as CLIP, visual re-ranking significantly improves performance. Using CLIP directly for retrieval results in an average mRR@1 of 15.4 across the five benchmarks. When using a multi-modal representation as described in Equation 1 of the main paper, the performance improves to an average mRR@1 of 29.4. Finally, when augmented with our visual re-ranking, the performance for mRR@1 boosts to 47.9. This confirms that the retrieval and visual re-ranking pipeline consistently improves performance over any single encoder with minimal overhead.

---

> ### Author Response · Authors · 2025-11-20
> **Rebuttal by Authors Part [2/2]**
>
> > **W3. The retrieval pool includes over 11,000 species—substantially larger than prior benchmarks—yet it remains unclear whether species appearing in test sets (e.g., CUB, iNat) are excluded from the retrieval corpus during evaluation. If not, this introduces a fairness issue, as the model could indirectly access prior knowledge of test species during inference.**
>
> We appreciate the reviewer’s concern, and we agree that ensuring a fair evaluation in an open-vocabulary setting is essential. To clarify, while our retrieval corpus includes all 11,202 species, by design, none of the test images from the benchmarks appear in our catalog. This ensures that the model never sees any test image during retrieval.
>
> Including all species names in the retrieval pool is intentional and follows the standard open-vocabulary and RAG formulation: the model must identify the correct species description from a large, fixed, non-parametric memory at inference time. Excluding test species from the retrieval pool would mean the retrieval module could never retrieve the correct description, effectively making open-vocabulary recognition impossible. This setup also mirrors prior work such as Retrieval-Augmented Classification [1], where the external memory includes all classes, even those used for evaluation.
>
> Importantly, all baselines use the same retrieval corpus, ensuring that no method receives privileged information. Performance differences arise solely from differences in retrieval quality, not from unequal access to species information.
>
> [1] Long, Alexander and Yin, Wei and Ajanthan, Thalaiyasingam and Nguyen, Vu and Purkait, Pulak and Garg, Ravi and Blair, Alan and Shen, Chunhua and van den Hengel, Anton. Retrieval Augmented Classification for Long-Tail Visual Recognition. CVPR 2022

---

> ### Comment · Reviewer_2xRi · 2025-11-25
>
> Regarding W2, I would question the claim that all species have common English names—this is likely not the case, especially for many species in iNaturalist, which contain only (Greek / Latin) scientific species names.
>
> A major concern that remains is the lack of clear benefits over simpler RAG baselines, primarily due to the absence of direct comparisons. Without this, it's difficult to assess the practical value of the proposed method.
>
> Follow-up to W3: What is the advantage of this pipeline over prior work such as [1], given that it follows a similar setup? While it's well-established that RAG can improve model performance, the paper lacks a direct comparison against a standard RAG baseline using the same data. That comparison is critical to isolate the specific contribution of this method. That said, I do appreciate the paper’s focus on enabling large-scale RAG for biology-specific applications, which is a valuable contribution—but the baseline comparison remains necessary to fully assess the method's impact.
>
> [1] Long, Alexander and Yin, Wei and Ajanthan, Thalaiyasingam and Nguyen, Vu and Purkait, Pulak and Garg, Ravi and Blair, Alan and Shen, Chunhua and van den Hengel, Anton. Retrieval Augmented Classification for Long-Tail Visual Recognition. CVPR 2022

---

> > ### Author Response · Authors · 2025-11-26
> >
> > Thank you for raising these points. Below, we have addressed the points individually.
> > ___
> > > **Q1: common name vs scientific names**
> >
> > As all the datasets are evaluated across the same label space of 11,202 species. We used common species names across all datasets to ensure consistency in the pipeline. For this, it is ensured that all the labels from the test datasets are mapped to their common names. iNaturalist provides both common and scientific names, we used the common names.
> >
> > > **Q2: Simpler Baseline comparison**
> >
> > We have shown that simpler baselines, such as CLIP, offer little to no improvement when used as the retrieval module, as seen in Table 3 (Qwen2.5+ Direct RAG). Direct RAG uses CLIP as a retrieval module. We also report the result below:
> >
> >
> > | Method | Average Accuracy |
> > |-----|------|
> > | DIRECT-RAG | 26.6 |
> > | VR-RAG| 55.2 |
> >
> > We also report the results with CLIP for other MLLMs and domains in Tables 10 and 11, respectively. In both cases, our approach shows significant gains over CLIP.
> >
> > > **Q3: Advantages of VR-RAG over RAC**
> >
> > The similarity in VR-RAG and RAC lies in that during evaluation, the test classes are present in the database index to be used by the retrieval. However, RAC outputs a probability distribution over the predefined label space, making it unsuitable for open-vocabulary settings. So, VR-RAG provides several key advantages compared to RAC:
> >
> > * VR-RAG supports Open-Vocabulary recognition and is training-free, while RAC is used for classification with a closed set of labels.
> > * RAC operates within a predefined label space and assumes the training distribution. In contrast, VR-RAG is more flexible and utilizes Wikipedia-derived summaries as its knowledge source, enabling the recognition of new categories.
> > * VR-RAG leverages summaries capturing fine-grained semantic traits (e.g., "red patch on the wing", "red tip on black-bill) that may be difficult for visual encoders to learn without explicit supervision. In contrast, RAC relies primarily on predefined class labels, without incorporating rich textual descriptions, making it difficult to adapt RAC to open-vocabulary scenarios.
> >
> > Overall, VR-RAG is especially effective in Open-World Recognition, where the number of categories is large, dynamic, or unbounded, and does not require any training. Newly discovered species can be integrated by adding their information to the catalog, making it a more scalable solution.
> > On the other hand, RAC produces a probability distribution over a predefined closed label space, not allowing for open-vocabulary recognition.

---

> > > ### Comment · Reviewer_2xRi · 2025-11-26
> > >
> > > Thanks for the clarifications, I have decided to bump up my score.

---

> > > > ### Author Response · Authors · 2025-11-26
> > > >
> > > > Thank you once again for your time and the thoughtful exchange, which has meaningfully improved the quality of our work.

---

### Official Review · Reviewer_BZEM · 2025-10-30

**Soundness:** 3
**Presentation:** 2
**Contribution:** 3
**Rating:** 6
**Confidence:** 4

**Summary:**

This paper presents VR-RAG (Visual Re-ranking Retrieval-Augmented Generation), a system for large-scale fine-grained species identification, focusing mainly on birds.
The authors argue that current closed-world benchmarks (e.g., CUB-200, NABirds) fail to reflect the real complexity of biodiversity recognition, where species are numerous, visually similar, and new species are continuously added.
VR-RAG combines three pretrained components:

Multimodal retrieval using CLIP / OpenCLIP / SigLIP to find candidate species descriptions from a curated Wikipedia-derived knowledge base (11 k+ species).

Visual re-ranking using DINOv2 to reorder retrieved candidates by pure image–image similarity, claimed to suppress environmental bias.

Reasoning with a pretrained MLLM (e.g., Qwen2.5-VL or InternVL) that receives the image and top-k candidate summaries and generates the final predicted species name.
The authors report consistent improvements in retrieval quality (mRR@k) and final classification accuracy across five bird datasets, FishNet, and Pokémon.

**Strengths:**

- Ambitious and environmentally meaningful problem framing: scaling biodiversity recognition to real-world, open-vocabulary conditions.

- Clear modular system design that combines retrieval, re-ranking, and reasoning.

- Demonstrated cross-domain generality (birds, fish, Pokémon).

- The curated textual summaries for >11 k species constitute a useful resource for the community.

**Weaknesses:**

- Scientific novelty – The system reuses existing pretrained encoders and an MLLM without new learning mechanisms. The main novelty is the composition, not a new algorithmic contribution.

- Prompt sensitivity – The MLLM reasoning stage relies on a single fixed prompt; no ablations on phrasing, candidate ordering, or number of candidates are shown. The robustness of the reported gains is therefore uncertain.

- Intra-species visual variability – Many bird species exhibit strong sexual dimorphism (male vs. female), seasonal plumage differences, or distinct juvenile appearances. The current setup appears to use one reference image and one textual summary per species. This could bias retrieval toward a single morph and hurt generalization. Please discuss or evaluate using multiple reference embeddings per species.

- Single-object assumption – Datasets used contain one main bird per frame; it is unclear how the framework handles multi-object or occluded scenes.

- Visual re-ranking explanation – The paper attributes improvement to DINOv2’s self-distillation producing object-centric features that suppress background noise. While plausible, this claim is unverified; no attention visualizations or controlled background tests are provided.

- Scope of contribution – Since no training is performed, the paper would benefit from clearly positioning itself as a system-level integration study rather than implying a new learning framework.

**Questions:**

- Include prompt ablations for the MLLM reasoning step.

- Discuss or extend to multi-reference per species to address male/female, juvenile, or seasonal variation.

- Provide visual evidence or controlled tests showing that DINOv2’s self-distilled embeddings indeed suppress environmental noise.

- Clarify scientific scope—system integration vs. methodological innovation.

---

> ### Author Response · Authors · 2025-11-20
> **Rebuttal by Authors Part [1/2]**
>
> We sincerely value your time and feedback. Below, we have carefully addressed your specific concerns and questions individually and updated the main paper to reflect the changes.
> ___
>
> > **W1. Scientific novelty – The system reuses existing pretrained encoders and an MLLM without new learning mechanisms. The main novelty is the composition, not a new algorithmic contribution.**
>
> We acknowledge that VR-RAG does not introduce new training algorithms. Our contribution lies in demonstrating the effectiveness of existing pretrained models when integrated in a carefully designed pipeline for fine-grained species recognition, which includes:
>
> * Curated species knowledge base: A large-scale, high-quality knowledge base covering 11,202 bird species, with summaries distilled to retain visually discriminative traits.
> * Species-focused retrieval: Indexing and retrieving from a large-scale, curated species knowledge base.
> * Visual re-ranking: Re-ranking retrieval results using vision models to resolve subtle differences between visually similar species.
> * RAG integration for multimodal reasoning: Feeding refined species summaries to MLLMs for zero-shot reasoning about species identity.
>
> Together, these components reveal new insights about the limitations and strengths of VLMs in open-vocabulary biodiversity recognition, which has not been systematically studied before. The novelty is thus practical and empirical, demonstrating what is possible in practice with current models, rather than algorithmic.
>
>
> > **W2. Prompt sensitivity – The MLLM reasoning stage relies on a single fixed prompt; no ablations on phrasing, candidate ordering, or number of candidates are shown. The robustness of the reported gains is therefore uncertain.**
>
>  We evaluated our approach on multiple prompts and reported the best results achieved. We have added an ablation study in section 6.5 to analyse the sensitivity to prompts. We also show the average results for all the prompts below across the five benchmarks.
>
> | Prompt | Accuracy |
> |-----|------|
> | P1 | 53.3 |
> | P2 | 53.3 |
> | P3 | 53.3 |
> | P4 | 53.7 |
> | P5 | 52.7 |
>
> The average accuracy across all prompts is 53.3, indicating that the method is relatively robust to variations in prompts.
>
> > **W3. Intra-species visual variability – Many bird species exhibit strong sexual dimorphism (male vs. female), seasonal plumage differences, or distinct juvenile appearances. The current setup appears to use one reference image and one textual summary per species. This could bias retrieval toward a single morph and hurt generalization. Please discuss or evaluate using multiple reference embeddings per species.**
>
> Thank you for highlighting this point. In our current setup, we already incorporate intra-species variability by using all available Wikipedia images for each species, which on average provides 3.2 images per species. These images naturally capture variation across sex, age, and seasonal plumage whenever such examples exist. In addition, our LLM-generated textual summaries often include separate descriptors for males, females, and juveniles, further reducing bias toward a single morph. Examples of this can be seen in Figure 15 of the appendix.
>
> That said, we agree that explicitly modeling intra-species variation is a meaningful consideration. While our current approach implicitly captures these differences through multiple images and detailed summaries, incorporating multiple reference embeddings per species by separating male, female, juvenile, or seasonal morphs when available could further strengthen retrieval in cases of strong dimorphism. Implementing this would require additional manual data curation, so we view it as a complementary refinement rather than a necessity for the current method. We have added a discussion on this idea in section K of the appendix.
>
> > **W4. Single-object assumption – Datasets used contain one main bird per frame; it is unclear how the framework handles multi-object or occluded scenes.**
>
> Our benchmarks assume a single dominant bird per image, which is consistent with standard practices in fine-grained recognition [1,2,3]. While extending the framework to handle multi-object scenes, e.g., via instance-level detection using bird-specific detectors, is an interesting direction for future work, it falls outside the scope of our current setup.
>
> [1] Reed, Scott and Akata, Zeynep and Lee, Honglak and Schiele, Bernt. Learning Deep Representations of Fine-Grained Visual Descriptions. CVPR 2016.
>
> [2] Akata, Zeynep and Reed, Scott and Walter, Daniel and Lee, Honglak and Schiele, Bernt. Evaluation of Output Embeddings for Fine-Grained Image Classification. CVPR 2015.
>
> [3] Zhang, Han and Xu, Tao and Elhoseiny, Mohamed and Huang, Xiaolei and Zhang, Shaoting and Elgammal, Ahmed and Metaxas, Dimitris. SPDA-CNN: Unifying Semantic Part Detection and Abstraction for Fine-Grained Recognition. CVPR 2016.

---

> ### Author Response · Authors · 2025-11-20
> **Rebuttal by Authors Part [2/2]**
>
> > **W5. Visual re-ranking explanation – The paper attributes improvement to DINOv2’s self-distillation producing object-centric features that suppress background noise. While plausible, this claim is unverified; no attention visualizations or controlled background tests are provided.**
>
> That is an interesting observation, however, we attribute the improvement from DinoV2 to what previous works like [1] have shown, that using DinoV2 improves performance over contrastive-learning-based visual encoders like CLIP and not to self-distillation producing object-centric features that suppress background noise. Moreover, in Table 13, we conduct an ablation over various vision encoders as our re-ranking module to further show that indeed DinoV2 outperforms all of them.
>
> [1] Tong, Shengbang and Liu, Zhuang and Zhai, Yuexiang and Ma, Yi and LeCun, Yann and Xie, Saining. Eyes Wide Shut? Exploring the Visual Shortcomings of Multimodal LLMs. CVPR 2024.
>
> > **W6. Scope of contribution – Since no training is performed, the paper would benefit from clearly positioning itself as a system-level integration study rather than implying a new learning framework.**
>
> We have revised our introduction and conclusion to reflect more on the practical considerations and impact of our work.
>
> Please note, our goal is to investigate how existing Vision-Language Models (VLMs) perform in truly open-vocabulary species recognition. Through a carefully designed pipeline, we show that even state-of-the-art models face substantial challenges when tasked with recognizing thousands of species from limited or evolving textual and visual information.
>
> Specifically, we contribute:
>
> * A large-scale species knowledge base covering 11,202 species with textual and visual information, enabling evaluation in a realistic open-vocabulary setting that goes far beyond the closed-set regimes typical of prior work.
> * An integrated pipeline combining retrieval, visual re-ranking, and multimodal reasoning that leverages both textual and visual information from Wikipedia. We show that this integration is substantially more effective than relying on text alone.
> * A comprehensive large-scale analysis of off-the-shelf components, evaluating their performance under extreme fine-grained and multiple domains that have not been systematically studied in the literature.
>
> The novelty of our work lies in how these components are combined, adapted, and rigorously evaluated, revealing new insights into the strengths and limitations of modern VLMs for real-world biodiversity recognition.

---

> > ### Author Response · Authors · 2025-11-27
> > **Thank you for your time. We look forward to your feedback on our response.**
> >
> > Dear Reviewer,
> >
> > As the discussion period comes to a close, we would like to express our sincere appreciation for your time and thoughtful evaluation. In our previous response, we carefully addressed your comments and provided detailed clarifications on the following aspects:
> >
> > * Clarifying the novelty and contributions of our work
> > * Analysing the Prompt Sensitivity
> > * Intra-species visual variability
> > * Single object assumption
> > * Visual re-ranking explanation
> > * Clarifying the scope of contribution
> >
> > We hope these clarifications have resolved your concerns. If you have any additional questions or points that require further clarification, we would be happy to elaborate.
> >
> > Thank you once again for your thoughtful review of our submission.

---

### Official Review · Reviewer_xe6N · 2025-11-01

**Soundness:** 3
**Presentation:** 2
**Contribution:** 2
**Rating:** 6
**Confidence:** 5

**Summary:**

- Proposes VR-RAG, a framework combining retrieval and reasoning for open-vocabulary species recognition.
- Connects encyclopedic text (Wikipedia) with visual cues for grounding unseen species.
- Builds a large benchmark of 11 k+ bird species with GPT-4o-refined summaries and curated anchor images.
- Designs a two-stage pipeline: multimodal retrieval, visual re-ranking, and MLLM-based reasoning. The method demonstrates strong cross-domain generalization across birds, fish, and Pokémon datasets

**Strengths:**

- The motivation of the paper is clear and significance.
- The authors construct a large-scale benchmark of more than 10k bird species and  produce concise, discriminative summaries aligned with visual evidence.
- The proposed VR-RAG framework demonstrated substantial accuracy gains over all baseline models across five bird datasets and two cross-domain settings.

**Weaknesses:**

- The idea of combines retrieval + re-ranking + MLLM reasoning is an intuitive and common way to bridge textual and visual knowledge bases. The contribution to novelty is limited.
- Wikipedia tends to emphasize well-studied taxa, resulting in regional and taxonomic biases that leave numerous cryptic or underrepresented species undocumented. How are species without Wikipedia pages handled? Is coverage gap quantified?
- Some design choice are not deeply analyzed, such as the number of retrieved candidates, or alternative retrieval architectures.

**Questions:**

- The paper mentioned the data curation using GPT-4o to filter. Have authors check whether this step will bring in bias and noise?
- The study focuses mainly on classification and retrieval accuracy, without exploring more demanding downstream tasks such as visual question answering.
- How much gain comes from re-ranking vs. GPT-4o summary quality alone?

---

> ### Author Response · Authors · 2025-11-20
> **Rebuttal by Authors Part [1/2]**
>
> We greatly value your time and feedback. Below, we have carefully addressed your specific concerns and questions individually. We have integrated changes into the main paper.
>
> ___
>
> > **W1. The idea of combines retrieval + re-ranking + MLLM reasoning is an intuitive and common way to bridge textual and visual knowledge bases. The contribution to novelty is limited.**
>
> We agree that retrieval + re-ranking + reasoning is conceptually intuitive. However, the novelty in our work lies in how these components are integrated and adapted specifically for large-scale, fine-grained species recognition.
>
> * The curation of Wikipedia summaries into visually salient descriptions enables MLLMs to reason effectively about subtle morphological differences, which is critical for species classification and not as effective as with entire Wikipedia articles(Table 7). Our approach is also different from prior generic RAG pipelines, which generally feed only the retrieved text chunk to the MLLM. We show in Table 7 that when only retrieved chunks are fed, the performance drops significantly. We also report the results from Table 7 below.
> * Our visual re-ranking explicitly addresses the challenge of fine-grained retrieval among highly similar species, a problem rarely considered in standard RAG systems.
> * We provide a comprehensive evaluation across 11,202 species and multiple benchmarks (birds, marine species, Pokémon), revealing practical insights into the behavior of modern VLMs under extreme fine-grained, long-tail, and cross-domain conditions.
>
> In other words, while each component individually exists, our careful adaptation, integration, and evaluation for species recognition constitute the technical contribution.
>
> | Model          | Birdsnap| CUB |  iNat | Ind Birds | NABirds | Average |
> | -------------- | ------- | --- | ----- | --------- | ------- | ------- |
> | Chunks         | 41.7    | 46.9| 26.0  |  52.1     | 47.2    | 42.8    |
> | Wikipedia      | 47.7    | 56.4| 33.6  |  65.8     | 52.7    | 51.2    |
> | Ours           | 51.9    | 60.3| 35.5  |  72.2     | 56.3    | 55.2    |
>
>
> > **W2. Wikipedia tends to emphasize well-studied taxa, resulting in regional and taxonomic biases that leave numerous cryptic or underrepresented species undocumented. How are species without Wikipedia pages handled? Is coverage gap quantified?**
>
> This is an interesting point. Wikipedia indeed exhibits regional and taxonomic biases, but for birds specifically, the coverage is effectively complete. We compare our species list with the IOC World Bird List v14.2 (11,250 globally recognized species) and find that Wikipedia contains entries for 11,202 bird species, slightly below the IOC count. So, species without Wikipedia pages are therefore excluded from our knowledge base.
> Also, for marine species, FishBase provides a rich source to extract information for all the marine species, with over 35,000 species listed.
>
> Additional sources like eBird, iNaturalist, or field guides can be used for extending our method to underdocumented species. We highlight this as an avenue for future work in section K of the appendix.
>
> > **W3. Some design choice are not deeply analyzed, such as the number of retrieved candidates, or alternative retrieval architectures.**
>
> We perform ablation on the number of retrieved candidates in Section 6.5: Top-K selection, where we show the effect of a larger retrieved pool. This experiment shows our approach is capable of handling a larger pool with minimal drop in performance.  We also report the results below.
>
> Additionally, we also study the effect of re-ranking at different thresholds in Section F of the appendix, showing that gains offered after 100 initial retrieved candidates are minimal.
>
> | Model          | Birdsnap| CUB |  iNat | Ind Birds | NABirds | Average |
> | -------------- | ------- | --- | ----- | --------- | ------- | ------- |
> | VR-RAG (k==5)  | 51.9    | 60.3| 35.5  |  72.2     | 56.3    | 55.2    |
> | VR-RAG (k==10) | 51.0    | 60.0| 35.2  |  71.3     | 56.1    | 54.7    |
> | VR-RAG (k==15) | 51.0    | 59.9| 35.0  |  70.3     | 56.1    | 54.5    |
>
>
> We focus on Vision Language pretrained CLIP-like architectures because they have consistently shown strong performance and are widely regarded as the standard approach for cross-modal retrieval[1]. For the re-ranking module, we experiment with several architectures, including CLIP, DinoV2, and Deit-III, and our results (Table 13) show that DINOv2 provides the strongest re-ranking performance.
>
> [1] Tianshi Wang and Fengling Li and Lei Zhu and Jingjing Li and Zheng Zhang and Heng Tao Shen. Cross-modal retrieval: a systematic review of methods and future directions, 2024

---

> ### Author Response · Authors · 2025-11-20
> **Rebuttal by Authors Part [2/2]**
>
> > **Q1. The paper mentioned the data curation using GPT-4o to filter. Have authors check whether this step will bring in bias and noise?**
>
> While completely unbiased and noise-free summaries cannot be guaranteed, we addressed this by performing two tests.
> * By manually validating 1,000 summaries against their corresponding bird images, ensuring they are useful for identifying the species. We also show some summaries in the appendix in section L.
> * We conduct a matching test, where the users are shown a description and five images, and they are asked to choose the image that is most matching. To ensure the task is fine-grained, we conducted this test on the 200 species from the cub dataset. We ensure the correct image is always present in the five images, and the remaining four are chosen as the closest to the description in the clip space. On average, users spent almost one minute on each description and achieved 94% accuracy. For the remaining 6%, 4% cases were where the users could not converge to one single option, and the remaining 2% were wrong. Qualitative samples for all the cases with provided explanation in section L.1. The summaries were found to be effective at capturing the visual attributes necessary for bird identification.
>
> > **Q2. The study focuses mainly on classification and retrieval accuracy, without exploring more demanding downstream tasks such as visual question answering.**
>
> Our work intentionally focuses on classification and retrieval core tasks in open-vocabulary, fine-grained species recognition, because these directly reflect real-world needs in biodiversity monitoring, ecological annotation, and expert-level identification. More demanding downstream tasks such as visual question answering (VQA) are indeed valuable, but they introduce challenges (e.g., linguistic compositionality, ambiguous queries) that are orthogonal to the central problem of species-level discrimination.
>
> The goal of this paper is to isolate and rigorously evaluate the species-recognition capability of vision-language models in an open-vocabulary setting. Our approach targets the primary bottleneck: fine-grained alignment between visual evidence and retrieved species descriptions. By grounding the model in multimodal species catalogs, our method encourages detailed reasoning over attributes such as beak color, wing pattern, or eye markings, thereby improving classification accuracy.
>
> That said, our framework is fully compatible with VQA-style pipelines. Extending our species-grounded retrieval system to question-answering tasks like “Which bird has the red crown?” or “Is this species migratory?” is a natural and promising direction for future work. We have added a discussion of this in Section K.
>
> > **Q3. How much gain comes from re-ranking vs. GPT-4o summary quality alone?**
>
> To illustrate the gains provided by re-ranking over the GPT‑4o summary, we removed the visual information from our pipeline. As a result, retrieval performance drops significantly, with an average 24.8% decrease on the mRR@1 metric across the five datasets, while recognition performance decreases by 18.0%. This suggests that removing visual information has a stronger impact on retrieval than on recognition, which can be attributed to the continued relevance of textual descriptions and the reasoning capabilities of the MLLM.

---

> > ### Author Response · Authors · 2025-11-27
> > **Thank you for your time. We look forward to your feedback on our response.**
> >
> > Dear Reviewer,
> >
> > As the discussion period approaches its end, we would like to extend our sincere gratitude for the time and thoughtful feedback you have provided. In our previous response, we carefully considered your comments and offered detailed clarifications on the following points:
> >
> > * Clarifying the contributions of our work
> > * Potential biases from Wikipedia and GPT-4o
> > * Design decisions behind VR-RAG
> > * Applicability to more challenging tasks
> > * Performance gains attributable to summary quality from GPT-4o
> >
> > We hope these explanations have addressed your concerns. Should you have any remaining questions or require further details, we would be glad to provide additional clarification.
> >
> > Thank you once again for your valuable review of our work.

---

### Official Review · Reviewer_PS7G · 2025-11-03

**Soundness:** 3
**Presentation:** 3
**Contribution:** 2
**Rating:** 2
**Confidence:** 4

**Summary:**

The paper presents a RAG-based framework for species classification that uses wikipedia summaries of species as the knowledge database. Firstly, an LLM is used to summarize the Wikipedia articles and then are divided into chunks. Multimodal embeddings of anchor images and the text chunks are created which are used to retrieve appropriate text chunks given a query image, which is embedded with a ensemble of VLMs. Finally, an MLLM is used to rerank the retrieved articles for final prediction. The proposed RAG-based system achieves state-of-the-art performance in various species classification benchmarks.

**Strengths:**

1. The paper proposes a RAG-based species classification pipeline using wikipedia article summaries and multimodal ensemble-based retrieval method.

**Weaknesses:**

1. Limited technical novelty as it is a simple RAG system that is used everywhere now but applied to species image classification. How is the framework different than just an application (RAG + x) where x is some problem? What are some exclusive properties of the proposed system that is tailored to species classification only? Can it be applied to other problems?
2. What is the computational overhead of the proposed RAG system as compared to the other VLMs. What is the cost-benefit ratio of using this system? What is the inference time given a query?
3. Is it possible to integrate additional modalities as done by recent works such as TaxaBind? Would incorporating additional modalities improve retrieval and the overall understanding capabilities of the model.
4. How robust is the proposed RAG framework to the database? Can the authors provide experiments for ablating the size of the database and robustness to noisy articles in the database?
5. Can you provide some uncertainty estimates to when reranking is not useful? What are the concrete failure cases when reranking is incorrect and retrieval is accurate?
6. In general, since this is in applications track, the paper needs to have more systems level analysis and experiments. I dont see particular systems level innovations.
7. All the experiments are provided on commonly used bird classification benchmarks. Any use cases of the RAG system beyond identification of common species? Like can you provide some experiments for novel species identification?

**Questions:**

Please see weaknesses.

---

> ### Author Response · Authors · 2025-11-20
> **Rebuttal by Authors Part [1/3]**
>
> We greatly appreciate your time and feedback. Below, we have carefully addressed your specific concerns and questions individually and integrated changes into the main paper.
>
> ___
> > **W1. Limited technical novelty as it is a simple RAG system that is used everywhere now but applied to species image classification. How is the framework different than just an application (RAG + x) where x is some problem? What are some exclusive properties of the proposed system that is tailored to species classification only? Can it be applied to other problems?**
>
> While RAG itself is a general paradigm, our framework introduces several species-specific adaptations that distinguish it from a straightforward application:
>
> * Curated species knowledge base: We build a large-scale, high-quality knowledge base covering 11,202 bird species, with summaries distilled to retain visually discriminative traits (e.g., wing color, leg color, beak shape). This curation is specifically designed to address the fine-grained visual distinctions inherent in species classification, which generic RAG applications would not handle effectively.
> * Evaluation at extreme scale: Our framework is validated on a large label space of 11,202 species, far beyond the limited classes typically used in previous works. This demonstrates robustness to large and fine-grained distributions.
> * Visual re-ranking tailored to species: We integrate a vision-based re-ranking module that exploits cues among visually similar species. This step is essential in fine-grained classification, where textual retrieval alone is insufficient, as shown in Table 3 of our paper, where CLIP achieves an average accuracy of 15.4, compared to ours 55.2.
> * Cross-domain applicability: We demonstrate this by applying VR-RAG to marine species (FishNet) and Pokémon datasets, showing the framework generalizes to other domains.
>
> Thus, our contribution lies in the data curation, design, adaptation, and empirical validation of a RAG-based system for open-vocabulary fine-grained species recognition.
>
>
> > **W2. What is the computational overhead of the proposed RAG system as compared to the other VLMs. What is the cost-benefit ratio of using this system? What is the inference time given a query?**
>
> We agree that practical applications must be computationally feasible. To address this, we have added a Systems Efficiency Analysis in Section 6.6. We compared the inference cost of our method against the standard baselines. As shown below, our method achieves substantial performance gains with only modest computational overhead:
>
> | Model              | mRR@1 (Avg.)   | Inference Time (s/img)|
> | :----------------- | :------------- | :----------------------|
> | VR-RAG             | 52.3           | 0.29s                  |
> | CLIP            | 15.4           | 0.17s                  |
> | OpenCLIP           | 13.0           | 0.16s                  |
> | SigLIP         | 18.5           | 0.16s                  |
>
> This analysis demonstrates that our pipeline is computationally practical for real-world deployment in biodiversity monitoring systems. Please refer to the main paper for more details.
>
>
> > **W3. Is it possible to integrate additional modalities as done by recent works such as TaxaBind? Would incorporating additional modalities improve retrieval and the overall understanding capabilities of the model.**
>
> Yes, new modalities can be integrated in a similar fashion to the visual modality. Works like ImageBind[1] and TaxaBind[2] have shown how multiple modalities can improve performance. It would be an interesting approach to try this, but collecting data at this scale(11,202 species) is a big bottleneck in being able to do so. We have added a discussion on promising future works in section K of the appendix, and about TaxaBind in the related work section.
>
> [1] Girdhar, Rohit and El-Nouby, Alaaeldin and Liu, Zhuang and Singh, Mannat and Alwala, Kalyan Vasudev and Joulin, Armand and Misra, Ishan. ImageBind: One Embedding Space To Bind Them All. CVPR, 2023
>
> [2] Srikumar Sastry and Subash Khanal and Aayush Dhakal and Adeel Ahmad and Nathan Jacobs. TaxaBind: A Unified Embedding Space for Ecological Applications. WACV 2025

---

> ### Author Response · Authors · 2025-11-20
> **Rebuttal by Authors Part [2/3]**
>
> > **W4. How robust is the proposed RAG framework to the database? Can the authors provide experiments for ablating the size of the database and robustness to noisy articles in the database?**
>
> Below, we assess the robustness of our RAG framework to the database by conducting an ablation in which retrieval is performed over the entire Wikipedia articles. However, please note that the refinement step, as described in section 3, is automated and achieved at a marginal price of 0.5 cents per catalog. Retrieval over refined summaries not only provides improved performance but is also much faster. A single query takes 0.29 seconds on our refined databases, while it takes 2.2 seconds over the noisy Wikipedia articles, as it is almost five times larger.
>
> The results of this experiment are included in Section 6.5 (Ablation Studies), we also report that below. The performance sees a small drop, but it remains largely stable despite the significantly noisier and larger database. This performance is still higher than VLMs like CLIP, OpenCLIP when performing retrieval over the clean database.
>
> | Model          |       | Birdsnap |        |       | CUB   |        |
> | -------------- | ----- | -------- | ------ | ----- | ----- | ------ |
> |                | mRR@1 | mRR@5    | mRR@10 | mRR@1 | mRR@5 | mRR@10 |
> | VR-RAG (noisy) | 47.1  | 48.6     | 49.0   | 56.8  | 58.5  | 59.0   |
> | VR-RAG (clean) | 48.9  | 52.6     | 53.8   | 58.0  | 62.3  | 63.1   |
>
> These findings show that even when exposed to large quantities of noisy text, the framework maintains strong retrieval performance, confirming its robustness to database size and noise.
>
> > **W5. Can you provide some uncertainty estimates to when reranking is not useful? What are the concrete failure cases when reranking is incorrect and retrieval is accurate?**
>
> We provide a detailed analysis in Appendix Section J, and summarize the key insights here.
>
> Our analysis focuses on two complementary scenarios: (1) cases where re-ranking harms the initial retrieval and (2) cases where it improves it.
>
> In failure cases, the mean similarity score shifts right after re-ranking. This indicates that many candidates receive similarly high scores, meaning the re-ranking model is confused because multiple species are visually very similar. In such cases, re-ranking, being sensitive to fine-grained appearance, tends to elevate additional hard negatives, causing the correct species to drop in rank as the difference between the correct and the other species' similarity is very small.
>
> In contrast, in success cases, the mean score shifts left, widening the gap between the true positives and the negatives. This indicates higher confidence: the top-5 retrieval set is indeed dominated by the correct or near-correct species, by the re-ranker.
>
> This effect of shifting is more pronounced in the CUB and NABirds datasets, which is shown in the figures in Section J. Using these observations, we derive a simple heuristic to detect when re-ranking is likely to be harmful. Applying this heuristic, on CUB mRR@10 shows a small increase from 63.1 to 63.8, and on NABirds from 57.6 to 58.4. Full analysis and details of the heuristic are provided in Appendix Section J. We also provide qualitative examples of failure and success cases in Section J.

---

> > ### Author Response · Authors · 2025-11-20
> > **Rebuttal by Authors Part [3/3]**
> >
> > > **W6. In general, since this is in applications track, the paper needs to have more systems level analysis and experiments. I dont see particular systems level innovations.**
> >
> > We agree that the Applications Track places emphasis on practical impact, empirical rigor, and real-world relevance. We have revised our introduction and conclusion to reflect more on these practical considerations and impact. We have also added an efficiency analysis in the updated version.
> >
> > In the meantime, based on the public ICLR guidelines, our understanding is that the Applications Track does not require systems-level innovations in the same way as ML Systems workshops. Instead, the stated criteria emphasize:
> >
> > * Novel applications of machine learning
> > * Significant empirical evaluation demonstrating real-world relevance
> > * Insights that broaden ML understanding in practical domains
> >
> > Our contribution aligns with these expectations.
> >
> > 1. Novel Application Domain: We introduce an open-vocabulary bird species recognition pipeline spanning 11,202 species, at a much larger scale than previous works.
> >
> > 2. Comprehensive System Construction: While we do not propose a new systems architecture, the work integrates retrieval, RAG-based reasoning, multimodal inference,  and large-scale data curation into a cohesive and deployable pipeline. We believe this constitutes system-level application engineering, consistent with other papers in the Applications Track.
> >
> > 3. Extensive Empirical Analysis:
> >
> >    * Large-scale retrieval experiments across 11,202 species
> >    * Open-vocabulary evaluation on five avian benchmarks, one marine benchmark, and one Pokémon benchmark.
> >    * Cross-model comparisons (CLIP, Qwen2.5-VL, InternVL-3)
> >
> >
> > > **W7. All the experiments are provided on commonly used bird classification benchmarks. Any use cases of the RAG system beyond identification of common species? Like can you provide some experiments for novel species identification?**
> >
> >
> > We respectfully disagree with the statement. In addition to standard bird benchmarks, we evaluate VR-RAG on two additional domains: Marine Species and Pokémon. The marine dataset contains over 17,000 species, including many rare and visually similar species, making it a significantly more challenging open-vocabulary recognition task. Our approach achieves substantial improvements over all baselines in this setting, demonstrating its effectiveness beyond common or well-studied categories.
> >
> > Moreover, our framework naturally supports species identification when a new species is discovered. It does not require retraining or architectural changes; instead, one can simply add the new species’ catalog entry to the retrieval database, and the system can recognize it. This makes our method well-suited for dynamic, continuously expanding catalogs, as commonly encountered in real-world biodiversity applications.

---

> > > ### Author Response · Authors · 2025-11-27
> > > **Thank you for your time. We look forward to your feedback on our response.**
> > >
> > > Dear Reviewer,
> > >
> > > As the discussion phase nears its conclusion, we would like to express our sincere appreciation for your time and thoughtful comments. In our previous response, we carefully reviewed your feedback and provided detailed clarifications regarding:
> > >
> > > * Clarifying the novelty of our work
> > > * Analyses on computational overhead
> > > * Integration of additional modalities
> > > * Robustness of VR-RAG
> > > * Analyses on uncertainty
> > > * Clarifications about the Application Track
> > > * Experiments on additional domains
> > >
> > > We hope our explanations have adequately addressed your concerns. If there are any remaining questions or points that would benefit from further clarification, we would be happy to elaborate.
> > >
> > > Thank you once again for your valuable assessment of our work.

---

### Author Response · Authors · 2025-12-02
**Summary of rebuttal process.**

Dear AC, SAC, and PC,

We appreciate the time you have taken to ensure a fair rebuttal process. To help streamline the process, we summarize the key points of our responses below.

During the post-rebuttal phase, Reviewers PS7G, xe6N, and BZEM did not respond before the portal was closed.

Reviewer 2xRi raised a follow-up question regarding the use of common names and additional baselines. We clarified that all datasets use common names and provided the requested comparisons. The reviewer expressed satisfaction with our explanation and raised their score toward acceptance.

**Key positive remarks:**

* *(xe6N, BZEM, 2xRi)*: Clear and significant motivation
* *(xe6N, BZEM)*: Valuable large-scale benchmark
* *(xe6N, 2xRi)*: Strong empirical results
* *(xe6N, BZEM)*: Cross-domain generalization

The reviewers’ concerns primarily involved requests for clarification or small-scale experiments rather than substantial revisions, indicating that the core methodology and results were already solid.

**Our responses to the concerns:**

* **Novelty and contribution (PS7G, xe6N, BZEM):**
  We clarified the distinctions from prior RAG-based approaches and demonstrated the superiority of our re-ranking strategy across all the benchmarks.

* **Computational overhead (PS7G):**
  We provided a detailed breakdown of inference time and performance. VR-RAG achieves a 2.8× improvement over the best baseline with only a 1.8× latency increase.

* **Robustness to noise, prompts, and retrieved candidates (PS7G, xe6N, BZEM):**
  We added experiments showing that our method is robust to noisy inputs, insensitive to prompt variations, and capable of handling a larger pool of retrieved candidates.

* **Application track confusion(PS7G):**
  We clarified that application-track submissions do not require system-level innovation, and our contribution aligns with the expectations of this track.

* **Data quality(xe6N):**
  We explained the human verification efforts taken to ensure the dataset used in our work is of high quality.


We appreciate the time and effort you have taken to evaluate our work.

Best regards,

Authors of Paper 6235

---

### Meta-Review · Area_Chair_PMEA · 2026-01-08

**Summary:**

This paper introduces VR-RAG, a framework for open-vocabulary species recognition (specifically birds) using a two-stage Retrieval-Augmented Generation approach. The system retrieves species descriptions from a Wikipedia-derived knowledge base of 11,202 species using multimodal embeddings (CLIP-like ensembles) and refines these via a visual re-ranker (DINOv2) before final reasoning by an MLLM.

The reviewers and meta-reviewer acknowledge that VR-RAG is a well-executed framework with significant practical utility, demonstrating a state-of-the-art 18.0% improvement in species recognition accuracy. However, the paper faces a strong recommendation for rejection based on a perceived lack of fundamental scientific discovery or technical insight. The primary consensus is that the system is a RAG pipeline that effectively composes existing off-the-shelf components (CLIP, DINOv2, and MLLMs) without introducing new learning mechanisms or algorithmic innovations. It may be better suited to other venues that emphasize application values.

**Reviewer Concerns:**

Reviewers have the following concerns that are addressed in rebuttal:

Top-K Limitations: The authors provided an ablation study showing that increasing the number of candidate species (K) can actually "overwhelm" and "hinder the model's reasoning".

Benchmark Fairness: Reviewer 2xRi’s concerns regarding the retrieval pool and common names were resolved, leading to a score increase, though they still acknowledged the lack of "clear benefits over simpler RAG baselines". The authors mentioned the CLIP baseline results and the significantly advantage over it. However I am not convinced CLIP retrieval qualifies as RAG. The simpler RAG baseline seems to be the QWEN-VL+direct RAG, which does show lower quality than the proposed.

Outstanding concern:

A major outstanding concern remains the lack of unique scientific insight. While the authors argued that their analysis of failure cases and the resulting "heuristic measure" for re-ranking (derived from the difference between maximum and median scores) constitute empirical insight, Reviewer PS7G remained unconvinced, noting that the pipeline lacks "exclusive properties" tailored specifically to species classification that could not be found in any other fine-grained recognition problem.

Also, the authors proposed segmenting species summaries into smaller text chunks as a contribution to help adhere to context limits. However, this chunking approach is a common, standard technique in text-based RAG systems. Its presentation as a major innovation—and its exclusion from the baselines used for comparison—further limits the perceived scientific contribution of the work.

**Reviewer Scores:**

Reviewer PS7G: 2 (initially Reject). Maintains that the work is an application of existing tools without sufficient domain-specific ML innovation.

Reviewer xe6N: 6 (initially Marginally Above). Appreciates the benchmark scale but acknowledges limited technical novelty.

Reviewer BZEM: 6 (initially  Marginally Above). Views the work as a solid system-level integration study but agrees there is no new learning mechanism.

Reviewer 2xRi: 6 (initially Marginally Below). Increased score after clarifications on domain-agnosticism and fairness, though noted the lack of benefits over simpler RAG baselines.

---

### Decision · Program_Chairs · 2026-01-26

Reject